

# Measurements of aerosol and CCN properties in the Mackenzie River delta (Canadian Arctic) during Spring-Summer transition in May 2014

Paul Herenz[1], Heike Wex[1], Silvia Henning[1], Thomas Bjerring Kristensen[1,*], Florian Rubach[2],
Anja Roth[2], Stephan Borrmann[2], Heiko Bozem[3], Hannes Schulz[4], and Frank Stratmann[1]

[1]Leibniz Institute for Tropospheric Research, Leipzig, Germany
[2]Particle Chemistry Department, Max Planck Institute for Chemistry, Mainz, Germany
[3]Institute for Atmospheric Physics, Johannes Gutenberg University, Mainz, Germany
[4]Alfred Wegener Institute, Helmholtz Center for Polar and Marine Research, Bremerhaven, Germany
[*]now at: Division of Nuclear Physics, Lund University, Box 118, Lund SE-22100, Sweden

*Correspondence to:* Frank Stratmann (stratmann@tropos.de)

**Abstract.** Within the framework of the RACEPAC (Radiation-Aerosol-Cloud Experiment in the Arctic Circle) project, the Arctic aerosol, arriving at a ground based station in Tuktoyaktuk (Mackenzie River delta area, Canada), was characterized during a period of 3 weeks in May 2014. The observations of basic meteorological parameters and particle number size distributions ($PNSDs$) were indicative for the rapid transition from Arctic spring to summer that took place during the measurement period. Two distinct types of air masses were found. One type were typical Arctic haze air masses, termed as spring-type air masses, characterized by a mono-modal $PNSD$ with a pronounced accumulation mode at sizes above $100\,\mathrm{nm}$. These air masses were observed during a period when back trajectories indicate an air mass origin in the north east of Canada. The other air mass type is characterized by a bi-modal $PNSD$ with a clear minimum around $90\,\mathrm{nm}$, and with an Aitken mode consisting of freshly formed aerosol particles. Back trajectories indicate that these air masses, termed as summer-type air masses, originated from the northern Pacific. Generally total particle number concentrations ($N_{CN}$) ranged from 20 to $500\,cm^{-3}$, while cloud condensation nuclei (CCN) number concentrations were found to cover a range between less than 10 up to $250\,cm^{-3}$ for a supersaturation ($SS$) between 0.1 and $0.7\,\%$. The hygroscopicity parameter $\kappa$ of the CCN was determined to be 0.23 on average and variations in $\kappa$ were largely attributed to measurement uncertainties.

Furthermore, simultaneous $PNSD$ measurements at the ground station and on the Polar 6 research aircraft were performed. We found a good agreement of ground based $PNSDs$ with those measured between 200 and $1200\,\mathrm{m}$. During two of the four overflights, particle number concentrations at $3000\,\mathrm{m}$ were found to be up to twenty times higher than those measured below $2000\,\mathrm{m}$, and for one of these two flights, $PNSDs$ measured above $2000\,\mathrm{m}$ showed a different shape than those measured at lower altitudes. This is indicative for long range transport from lower latitudes into the Arctic that can advect aerosol from different regions in different heights.



# 1   Introduction

The Arctic region is particularly sensitive to climate forcing and reacts with amplifying feedbacks (e.g. the albedo-sea ice feedback) (Law and Stohl, 2007). Aerosol particles have the ability to modify these feedbacks in different ways. Black carbon depositions on ice surfaces can significantly change the surface albedo, supporting the uptake of heat due to absorption (Keegan et al., 2014). But also a change in the amount of available aerosol particles can modify cloud properties and indirectly change the energy budget. Twomey (1974) found that an increased number concentration of Cloud Condensation Nuclei (CCN) leads to smaller but more numerous cloud droplets if the same amount of water vapor is available for cloud formation. This can change the interaction with incoming shortwave radiation, increasing cloud reflectivity and therefore cooling. This effect might be of limited relevance in the Arctic since its surface is highly reflective due to snow and ice (Tietze et al., 2011). But a higher droplet number concentration would also lead to an increased cloud longwave emissivity which warms the Earth's surface (Garrett et al., 2002; Lubin and Vogelmann, 2006). Since low altitude clouds tend to be warmer than the Arctic surface, the Arctic is potentiality sensitive to this effect (Garrett et al., 2002). A shift of the droplet size distribution to smaller sizes can also affect cloud lifetime due to a possible later onset of precipitation (Albrecht, 1989). Hence the resulting issues that were in the focus of Arctic aerosol studies in the last years are the characterization of particle sources, their chemical and physical properties as well as the direct and indirect impacts of Arctic aerosol particles and pollutants on cloud forming properties (Jacob et al., 2010).

It is well known that the origin of Arctic air masses is dependent on the season. In winter and spring the Arctic aerosol is dominated by long-range transport of mid-latitude air masses. The polar front is located further south in areas of high anthropogenic pollution so that anthropogenic industrial emissions reach the Arctic atmosphere (Iversen and Joranger, 1985). Also biomass burning in Russia contributes to the high aerosol particle loading during winter and spring (Warneke et al., 2009). During polar night, the Arctic atmosphere is extremely stable which prevents turbulent mixing between vertical layers and with that also cloud formation and precipitation (Shaw, 1981). Hence the so called Arctic haze can be trapped for 15 up to 30 days (Shaw, 1981, 1995). The major part of the Arctic haze consists of particulate organic matter (POM) and sulfate but also contains ammonium, nitrate, mineral dust, black carbon and heavy metals (Quinn et al., 2002). During the transition from spring to summer an increased vertical mixing causes the presence of low-level clouds, and the related wet removal stops the Arctic haze period (Tunved et al., 2013). During the Arctic summer locally and freshly produced aerosol particle species [e.g. oxidation products of dimethylsulfide (DMS) (Quinn et al., 2007)] are dominant, caused by an increase in both biological activity and photochemistry (Ström et al., 2009). This results in a peak in new particle formation and a higher number concentration of aerosol particles in the Aitken mode range, whereas the well aged (Heintzenberg, 1980) Arctic haze particles of the accumulation mode disappear (Engvall et al., 2008; Tunved et al., 2013). Consequently, the Arctic aerosol particle number size distribution as well as the particle number concentration show a large seasonal variability (Tunved et al., 2013). Moreover the sources and sinks for Arctic aerosol particles are subject to the fast changes in the Arctic that currently take place. Dall'Osto et al. (2017) for instance found a negative correlation between the Arctic sea ice extent and new particle formation (NPF) events, that were observed at Mt Zeppelin (Svalbard). From this connection follows an increased new particle production due





to the current decrease in the sea ice pack extent (Dall'Osto et al., 2017).

Similarly, also the Arctic CCN number concentrations vary, with values between less than $100\,cm^{-3}$ (pristine Arctic background), occasionally less than $1\,cm^{-3}$ (Mauritsen et al., 2011), and up to $1000\,cm^{-3}$ (in Arctic haze layers, Moore et al. (2011) and references therein). In the previously mentioned study by Dall'Osto et al. (2017) it is also shown that the NPF

events and the growth of these aerosol particles to a larger size can affect the CCN number concentration. Dall'Osto et al. (2017) found an increase of the CCN number concentration (measured at a super saturation of 0.4 %) of 21 % which is linked to NPF events. Beside the fact that aerosol particles need to have a certain size to act as CCN, also the aerosol particle chemistry matters in terms of the activation to a cloud droplet. The single hygroscopicity parameter $\kappa$ (Petters and Kreidenweis, 2007) is commonly used to express the affinity of aerosol particles to water and characterizes their CCN activity. The hygroscopicity of

the Arctic aerosol particulate (PM) matter was also found to show a seasonality. $\kappa$ values determined from CCN measurements done on water soluble particulate matter collected in Spitzbergen by Silvergren et al. (2014) were between 0.3 and 0.7, with a minimum from March to May and a maximum in October. The past and future changes in the Arctic climate are related to changes of CCN number concentrations and their properties and consequently also to the sources and sinks of Arctic CCN. Hence there is a need for CCN measurements in the Arctic region to quantify these parameters. Furthermore, there is a lack of

in-situ measurements due to the seclusion of the Arctic region.

The data set presented in this study was recorded during the RACEPAC (Radiation-Aerosol-Cloud Experiment in the Arctic Circle) project, which took place in Inuvik (Canada) during April and May 2014. It was mostly an airborne campaign that aimed to measure all components required to describe the interaction of aerosol particles, clouds and radiation in the Arctic. In this framework an additional ground based station in Tuktoyaktuk ($\approx130\,km$ north of Inuvik) was installed and operated

by the Max Planck Institute (MPI) for Chemistry to measure Arctic CCN and aerosol properties. The data set presented here contains concentrations for condensation nuclei (CN) as well as for CCN, particle number size distributions ($PNSD$) and inferred particle hygroscopicity values ($\kappa$) measured at the station in Tuktoyaktuk. Further a comparison of $PNSD$s measured at the ground based station and on the research airplane Polar 6, operated by the Alfred Wegener Institute for Polar and Marine Research (AWI, Germany), is presented.

## 2  Experimental procedure and methods

### 2.1  Measurement setup and data processing

The experimental setup used for this study is shown in Figure 1. An aerosol inlet with a $PM_{10}$ sampling head was installed on top of a measurement container at a height of 3.5 meter above ground level. Along a vertical tube (inner diameter of 2.5 cm) the aerosol was transported into the measurement container. Downstream horizontal tubes (inner diameter of 1 cm and 0.53 cm)

distributed the aerosol to the instruments.

The total aerosol particle number concentration ($N_{CN}$) was measured by a Condensation Particle Counter (CPC, TSI Model 3010) which was operated at a total flow rate of 1 lpm. The CPC Model 3010 counts single aerosol particles between 10 nm (50 % of particles at this size are detected) and 3 µm. In parallel the particle number size distribution was measured by means



of a Scanning Mobility Particle Sizer (SMPS, TSI model 3936 with CPC Model 3025). The SMPS scanned aerosol particle mobility diameters from 13.6 nm up to 736.5 nm with a time resolution of 5 min. Upstream of the SMPS an impactor with 1 µm cutoff diameter was installed and prior to the CPC and the SMPS the aerosol was dried using a diffusion dryer with silica gel. The airborne measurements of $PNSD$s on board of the Polar 6 research aircraft were conducted by means of an

Ultra-High Sensitivity Aerosol Spectrometer (UHSAS) that measured in a size range from 70 nm up to 1 µm.

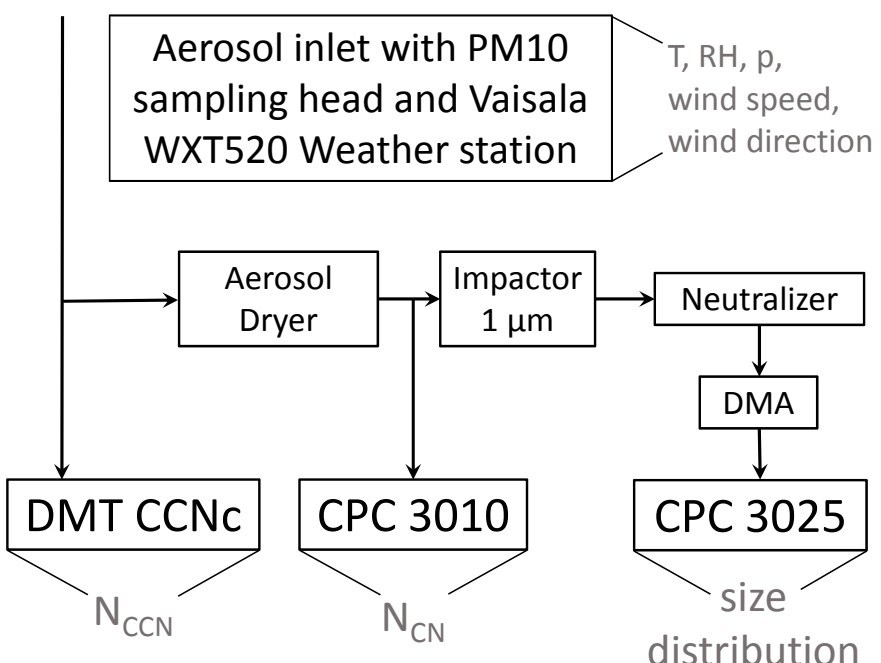

**Figure 1.** Schematic depiction of the measuring set-up implemented in the Tuktoyaktuk ground based (container) station.

The number concentration of cloud condensation nuclei ($N_{CCN}$) was measured using a Cloud Condensation Nuclei counter (CCNc, Droplet Measurement Technologies (DMT), Boulder, USA). The CCNc is a continuous-flow thermal-gradient diffusion chamber which is described in detail in Roberts and Nenes (2005). The CCNc was operated as recommended in Gysel and

Stratmann (2013) for polydisperse CCN measurements. By stepping the temperature gradient every 12 minutes the supersaturation ($SS$) was varied between 0.1 %, 0.2 %, 0.3 %, 0.5 % and 0.7 % at a constant total flow rate of 0.5 lpm. For consistency checks between $N_{CN}$ and $N_{CCN}$ also 1 % $SS$ was applied (not applied for analysis). To ensure stable column temperatures, the first 5 min and the last 30 sec at each $SS$ setting were excluded from the data analysis. The remaining data points were averaged. A $SS$ calibration of the CCNc had been done at the cloud laboratory of the Leibniz Institute for Tropospheric Re-

search (TROPOS) prior to the campaign, to determine the relationship between the temperature gradient along the column and



the effective $SS$. Based on recommendations given in Gysel and Stratmann (2013) and Rose et al. (2008) ammonium sulfate particles were size selected using a differential mobility analyzer (DMA type Hauke medium) and then fed into a CPC (TSI model 3010) and the CCNc which were operated in parallel. This was done for all $SS$ values that were also applied during the measurement campaign. A size dependent activated fraction was obtained by dividing $N_{CCN}$ obtained at different sizes by the respective $N_{CN}$. By fitting the resulting curve using a sigmoid function, the critical diameter $d_{crit}$ (where 50 % of all singly particles are activated) was determined. Applying the Köhler theory (Köhler, 1936) these $d_{crit}$ were used to determine the effective $SS$ based on the theoretical activation diameter of ammonium sulfate particles. The resulting effective $SS$ values for this study are 0.11 %, 0.21 %, 0.31 %, 0.51 % and 0.70 %, respectively. These calibrated values were used for further calculations, while the values reported from here on in the text are rounded values.

## 2.2 Inferring particle hygroscopicity

Whether an aerosol particle acts as a CCN depends on its size, chemical composition and the water vapor saturation in its vicinity. Köhler-theory can be used to model the critical saturation ratio $S_{crit}$ required for activation of a particle to a droplet (Köhler, 1936). The water activity, one term entering this theory, can be expressed based on a single parameter representation, using the hygroscopicity parameter $\kappa$ (Petters and Kreidenweis, 2007). The $\kappa$ values reported in this study were calculated as follows, assuming the surface tension of the examined solution droplets ($\sigma_{s/a}$) to be that of pure water:

$$\kappa = \frac{4A^3}{27d_{crit}^3 \ln^2 S} \tag{1}$$

with

$$A = \frac{4\sigma_{s/a} M_w}{RT\rho_w}. \tag{2}$$

$d_{crit}$ is the critical diameter at which particles are just large enough to be activated to a droplet when exposed to a certain $S$, the saturation ratio. $M_w$ and $\rho_w$ are the molar mass and density of water while $R$ and $T$ are the ideal gas constant and the absolute temperature, respectively. To derive $d_{crit}$, simultaneously measured $N_{CCN}$ and $PNSD$s were used. It was assumed that all particles in a neighborhood around a given relevant aerosol particle diameter have a similar $\kappa$, i.e., that the aerosol particles are internally mixed. From that it follows that at a given $SS$ all particles become activated to droplets when their dry size is similar to or larger than $d_{crit}$. Hence $d_{crit}$ is the diameter at which $N_{CCN}$ is equal to the value of the cumulative $PNSD$, with the integration being done from the largest measured diameter of the $PNSD$ to lower diameters. The thus derived $d_{crit}$ can then be used, together with the corresponding $S$ (i.e., the calibrated $SS$ at which $N_{CCN}$ was measured) to derive $\kappa$ for the ambient particles, based on Eq. 1. The inferred $\kappa$ values correspond to particles with sizes of roughly $d_{crit}$. The uncertainty in $\kappa$ which results from the uncertainties of the $PNSD$ measurements and the $SS$ of the CCNc, was determined by applying a Monte Carlo simulation in a similar fashion as done by (Kristensen et al., 2016). A detailed description of this method is given in Appendix A2.





For atmospheric particles, $\kappa$ can range between almost 0 for insoluble (e.g. soot and some organics) and 1.4 for very hygroscopic (e.g. sodium chloride) particles (Petters and Kreidenweis, 2007). A generally good estimate for a continental $\kappa$ of around 0.3 is given by Andreae and Rosenfeld (2008). Wex et al. (2010) report that the hygroscopicity of marine aerosol particles cover a broad range from several $\kappa$ values below 0.1 up to fewer values of 1, with a dominating $\kappa$ value of 0.45.

## 2.3   Measuring site and meteorology

All measurements presented in this study were performed during a period from May 1 to May 17 in 2014 at the outskirts of Tuktoyaktuk, a town of less than 1000 inhabitants in the Inuvik Region of the Northwest Territories, Canada. Figure 2 shows a map of Alaska and the western part of Canada together with the Sea Ice extent layer and the Corrected Reflectance layer of MODIS (Moderate Resolution Imaging Spectroradiometer). Tuktoyaktuk is situated north of the Arctic circle (69° 26' N and 133° 01' W) on the shore of the Beaufort Sea and 5 m above sea level. The area around Tuktoyaktuk has a low population density. The nearest town with more than 1000 inhabitants is 130 km to the south (Inuvik). The Beaufort Sea is located directly to the north of Tuktoyaktuk, and it is typically covered with ice from October to June. The pink color in Figure 2 shows the extent of the sea ice on the 15th of May 2014. The area of the frozen sea surface covers the entire Beaufort Sea and extends to the south to the Bering strait. Hence, it can be excluded that aerosol particles of marine origin are formed locally during the measuring period. The landscape surrounding the measurement station is comprised by flat Arctic tundra with a subarctic climate. The characteristic precipitation is less than 300 mm per year and the mean annual surface temperature is below 0 °C. The time series of the meteorological parameters temperature, relative humidity, pressure, wind speed and wind direction (Figure 3) give an overview with respect to the ambient weather conditions during the whole sampling period. The measured temperature shows an increasing trend typical for the transition from Arctic spring to summer. This transition is driven by the increase of the daily incoming solar radiation and leads to a change in sea ice and snow cover and consequently also to a change of Arctic aerosol particle sources. During this transition from spring to summer, the polar front is moving towards the north resulting in a more frequent passage of low pressure systems as well as enhanced dynamics and mixing in the boundary layer of subarctic areas. This can be seen in the high variability of the measured temperature (from −10 °C up to 15 °C) and the relative humidity (from 45 % up to 95 %). Furthermore, sharp changes in all meteorological parameters indicate the passage of local low pressure frontal systems (e.g. the pressure drop and the wind shift at the 13th of May that correspond to a cold front. Cold and warm fronts are marked with blue and red arrows in Figure 3, respectively) indicating a fast air mass change. In Figure 2 a low pressure system , which was located over the Beaufort Sea, is visible due to the Corrected Reflectance layer. The corresponding warm front is also visible in the meteorological parameters of Figure 3.





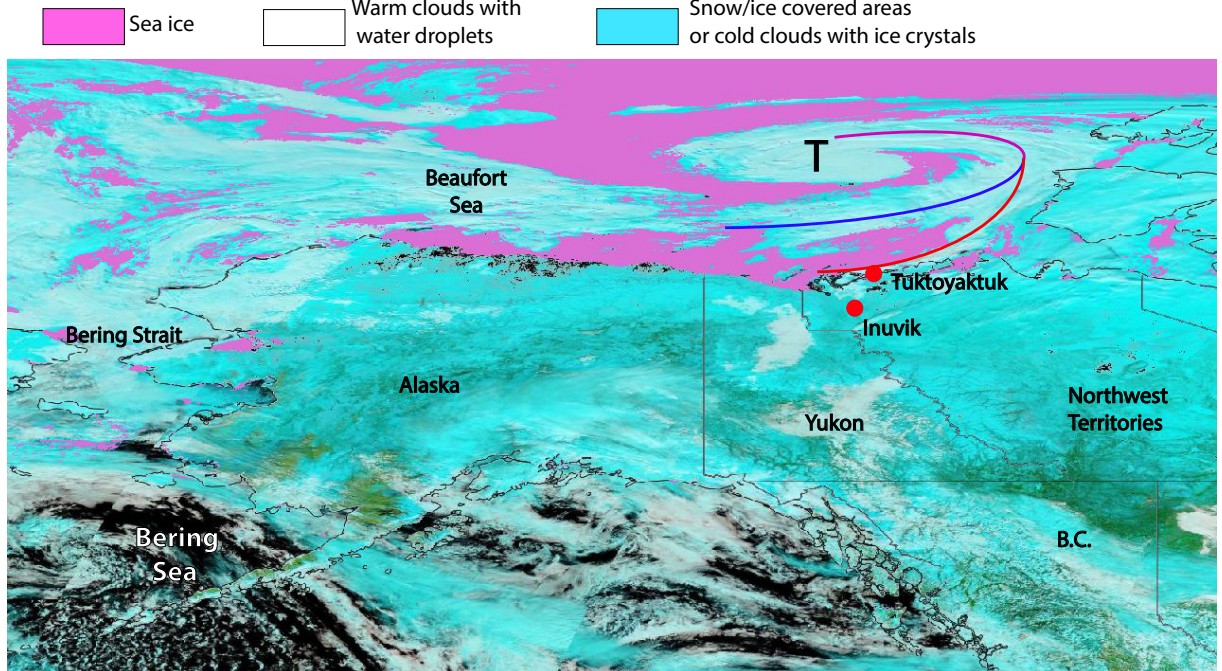

**Figure 2.** Map of Alaska and the western part of Canada showing the position of Tuktoyaktuk and Inuvik together with two additional layers of the MODIS instrument (installed on TERRA, NASA). The Sea Ice extent layer shows the frozen ocean surface in pink. The Corrected Reflectance (Bands 7,2,1) layer shows liquid water in dark blue or black. Ice on the surface or in form of ice crystals in cold clouds will appear turquoise whereas small water droplets in warm clouds will appear white. A low pressure system, which was relevant for the measurements in Tuktoyaktuk, and its corresponding fronts are marked. The map was created for the 15th of May 2014 using NASA Worldview (https://worldview.earthdata.nasa.gov/?map=-126.907471,36.373535,-117.415283,42.815918&products=baselayers, MODIS_Aqua_CorrectedReflectance_TrueColor~overlays,MODIS_Fires_All,sedac_bound&time=2012-08-23&switch=geographic).





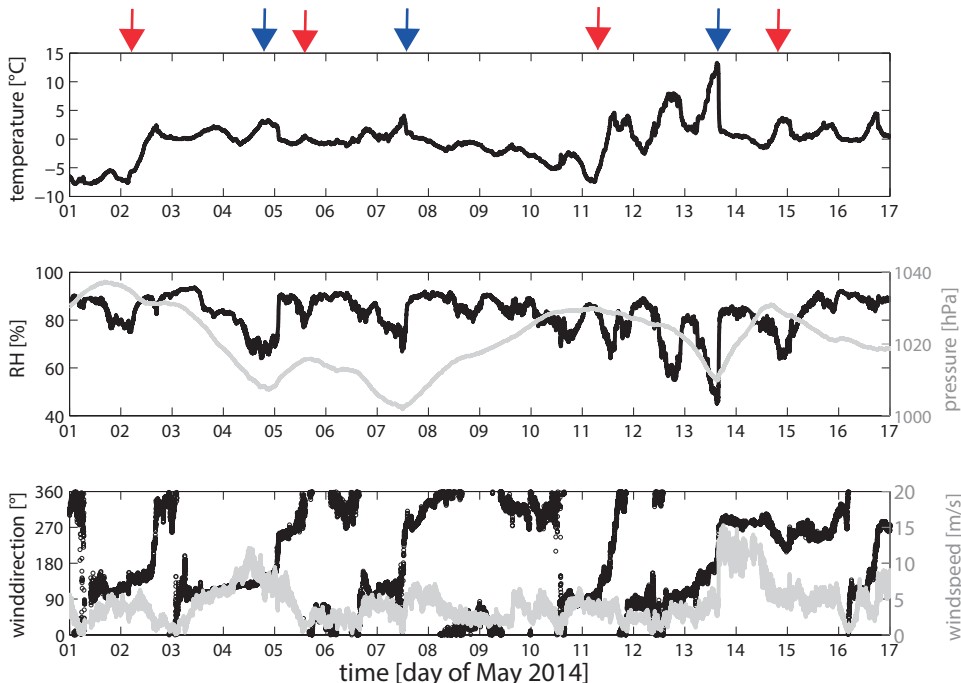

**Figure 3.** Time series of temperature, pressure, relative humidity as well as wind speed and direction measured directly at the aerosol inlet at an altitude of 3.5 m above ground level covering the entire measurement period in May 2014. The passages of cold and warm fronts are marked with blue and red arrows, respectively.

## 3   Results and discussion

### 3.1   Overview of $N_{CN}$, $N_{CCN}$ and $PNSD$ data for the entire measurement period

The measurements of all aerosol parameters recorded for this study were temporarily influenced by local anthropogenic sources. Local emissions from ground based sources such as industrial combustion, oil heating and the generator for the

5   container power as well as occasional air traffic led to intensive short term peaks in the measured time series of $N_{CN}$, $N_{CCN}$ at $SS$ above $0.1\,\%$ and the $PNSDs$.

The gray line in Figure 4 shows the raw time series of the measured total aerosol particle number concentration, where we will refer to these data using the parameter $N_{CNraw}$. Beside a clear baseline of concentrations between less than 100 and $1000\,cm^{-3}$, peaks up to more than $10000\,cm^{-3}$ occur during the whole sampling period. These peaks had a typical temporal

10   duration of 1 to 5 minutes. Consequently, the first step of the data analysis was the application of a filter routine to eliminate time periods were the measurements were affected by local pollution. The filtering procedure is described in Appendix A1. The black dots in Figure 4 are the remaining $N_{CN}$ data points. Note that especially during phases of high ambient pressure the local



pollution is more intensive and the filter eliminates most of the data during these periods. Hence, on the 2nd, 11th to 13th and on the 15th of May almost no $N_{CN}$ data points remain. These are time periods that directly follow the maxima in the pressure time series of Figure 3. Typical for high pressure systems are temperature inversions near the ground level that can trap local emissions and cause an enhanced influence of local pollution. Furthermore, Figure 4 shows $N_{CCN}$ measured at $SS = 0.1\%$

($N_{CCN,0.1}$) and number concentrations of particles larger than $150\,nm$ (integrated $PNSD$, $N_{CNraw>150nm}$) for the entire measurement period. For $N_{CNraw>150nm}$ and $N_{CCN,0.1}$ the filter procedure was not applied. Generally, $N_{CNraw>150nm}$ and $N_{CCN,0.1}$ show similar trends, and both do not show pronounced peaks as those seen for $N_{CNraw}$. This indicates that the observed aerosol particles that we related to pollution occurred in the size range below $150\,nm$. We will elaborate on this below. It is, however, also worth noting that $N_{CNraw>150nm}$ scatters much more than $N_{CCN,0.1}$. This originates in the higher frequency

with which the former was measured, i.e., it is associated with lower sampling statistics. Over the whole measurement period a mean $N_{CN}$ of $188\,cm^{-3}$ (and a median of $199\,cm^{-3}$) was observed when excluding the pollution periods. This is in good agreement with an Arctic long term study by Tunved et al. (2013) who report monthly mean $N_{CN}$ for May (observed at Mt Zeppelin, Ny-Alesund, Svalbard from March 2000 to March 2010) being slightly above $200\,cm^{-3}$. Generally they observed the highest concentrations between April and July which can be traced back to aged anthropogenic Arctic haze aerosol earlier

in this time period and to new particle formation later (Tunved et al., 2013). That these two kinds of aerosol also play a major role in context of the present study is discussed in the following two sections by using air mass back trajectories and the $PNSD$s.

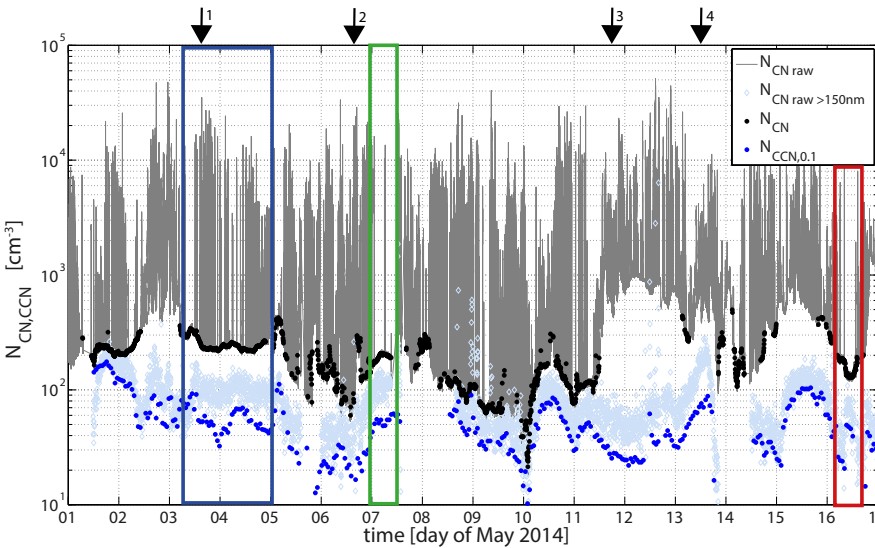

**Figure 4.** Time series of $N_{CNraw}$ (gray) and $N_{CN}$ (i.e., the filtered data, black) as well as $N_{CNraw>150nm}$ (light blue) and $N_{CCN,0.1}$ (dark blue). The colored boxes mark the three periods of measurements that were used for further analysis. The arrows at the top indicate the four overflight times of the Polar 6 research aircraft.


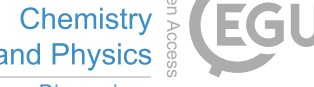

Figure 5 shows a comparison of the mean of all of those $PNSD$s which were detected during time periods that were marked as clean ($PNSD_c$, corresponding to times when $N_{CN}$ is shown as black dots in Figure 4) with the mean of all other $PNSD$s. The latter are those for which an influence of local pollution was assumed ($PNSD_p$). Also shown in Figure 5 are error bars that indicate the range between the 25 % and 75 % percentiles. Both, $PNSD_c$ and $PNSD_p$ are bimodal with an Aitken mode

5  below 100 nm and an accumulation mode above 100 nm. Above 150 nm, the accumulation modes of both are almost equal, whereas the Aitken mode of $PNSD_p$ is more pronounced than that of $PNSD_c$. Similar influences of local emission on $PNSD$ were found at an urban background station in Helsinki by Wegner et al. (2012). They observed modes between 10 nm and 40 nm in median urban $PNSD$ caused by traffic, domestic and district heating, comparable to our result, albeit at higher concentrations. This corroborates the assumption made earlier that the observed high peaks in $N_{CNraw}$ originate from local

10  pollution. It also demonstrates the usefulness of the applied filter (see Appendix A1). As for $N_{CN}$, also $PNSD_c$ agrees well with the observations of Tunved et al. (2013). A $PNSD$ shown in Tunved et al. (2013), representing the monthly median $PNSD$ for May over a period of ten years in an Arctic environment, shows the same characteristic as $PNSD_c$ of this study as shown in Figure 5. Both are bimodal with a distinct accumulation mode and a smaller Aitken mode. The variability of $PNSD_c$ and the dependence on the air mass origin is discussed in Section 3.3.

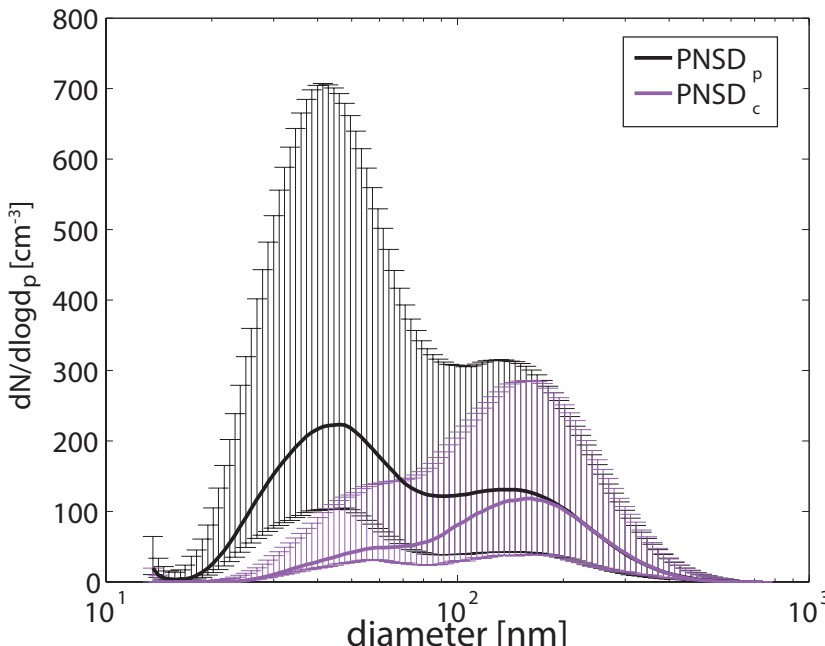

**Figure 5.** Comparison of the median $PNSD$s for times which were assumed to be clean ($PNSD_c$, purple) or polluted ($PNSD_p$, black). The thin vertical lines (same color code) indicate the range between the 25 % and 75 % percentiles.





After applying the filter on $N_{CCN}$ and the $PNSD$s only three distinct periods (Period 1: May 3 7:00 to May 5 0:00 , Period 2: May 7 00:10 to May 7 9:00, Period 3: May 16 5:00 to May 16 12:15) of evaluable data remained, as these parameters were measured with a lower temporal resolution and were thus more prone to be effected by local pollution than $N_{CN}$. The three periods are marked with a blue, green and red square in Figure 4. This color code is continuously used in the following figures.

The arrows at the top of Figure 4 indicate the four overflights of the Polar 6 research aircraft. For these times a comparison of ground based and airborne $PNSD$s of different altitudes was done, which is discussed in Section 3.5.

### 3.2 Origin of sampled air masses of the three periods

To assess the origin of the air masses which were investigated in Tuktoyaktuk we used the Lagrangian analysis tool (Sprenger and Wernli, 2015). The LAGRANTO backward trajectories were calculated based on analysis data from the European Center

of Medium-Range Weather Forecasts (ECMWF). The data used have a horizontal grid spacing of $0.5\,°$ and 137 vertical hybrid sigma-pressure levels from the surface up to $0.01\,\mathrm{hPa}$. Hourly 10-day back trajectories were started in the region around Tuktoyaktuk (69.43°N, 133.00°W) at a pressure level of $25\,\mathrm{hPa}$ below surface pressure. More specifically, we initialized trajectories at 13 receptor sites in the horizontal plane, accounting for the uncertainty introduced due to the relatively coarse horizontal grid and the release at an individual point. One receptor site was directly at the coordinates of the measurement

station and 12 around the station. The upper panel of Figure 6 depicts the bundle of trajectories for the three time periods. Two main air mass origins were observed. The air masses of Period 2 originated in the north east of Canada (in the province Nunavut). During May air masses from this area are typically highly contaminated due to high winter and springtime aerosol particle burdens which can be observed all over the Arctic (Shaw, 1995). In the following these air masses are termed "spring-type air mass". The air masses of Period 3 originated in the southwest of Tuktoyaktuk (Eastern Russia, Kamchatka and the

unfrozen North Pacific). In the following, these air masses are named "summer-type air mass". Further the trajectories indicate that Period 1 includes both, spring- and summer-type air masses. The distribution of these two air masses during Period 1 is visible in the lower panel of Figure 6. The figure shows the number of trajectories per hour (for Period 1) that originated east ($N_{spring-type}$) or west ($N_{summer-type}$) of Tuktoyaktuk. It can be seen that during the first part of Period 1 (till 16:00 of 04 May 2014) the air masses in Tuktoyaktuk were a mixture of spring- and summer-type air masses. Until the 3rd of May Tuktoyaktuk

was influenced by an anticyclone with a maximum pressure of $1035\,\mathrm{hPa}$. The low pressure gradient of this anticyclone led to a low wind velocity and a baffling wind (see lower panel of Figure 3) which caused an alternation between the two air mass origins at the beginning of Period 1. The second part of Period 1 is characterized by a decreasing surface pressure and a constant easterly wind. With a temporal shift of less than one day also $N_{summer-type}$ decreased which indicates that only spring-type air masses were present at Tuktoyaktuk.





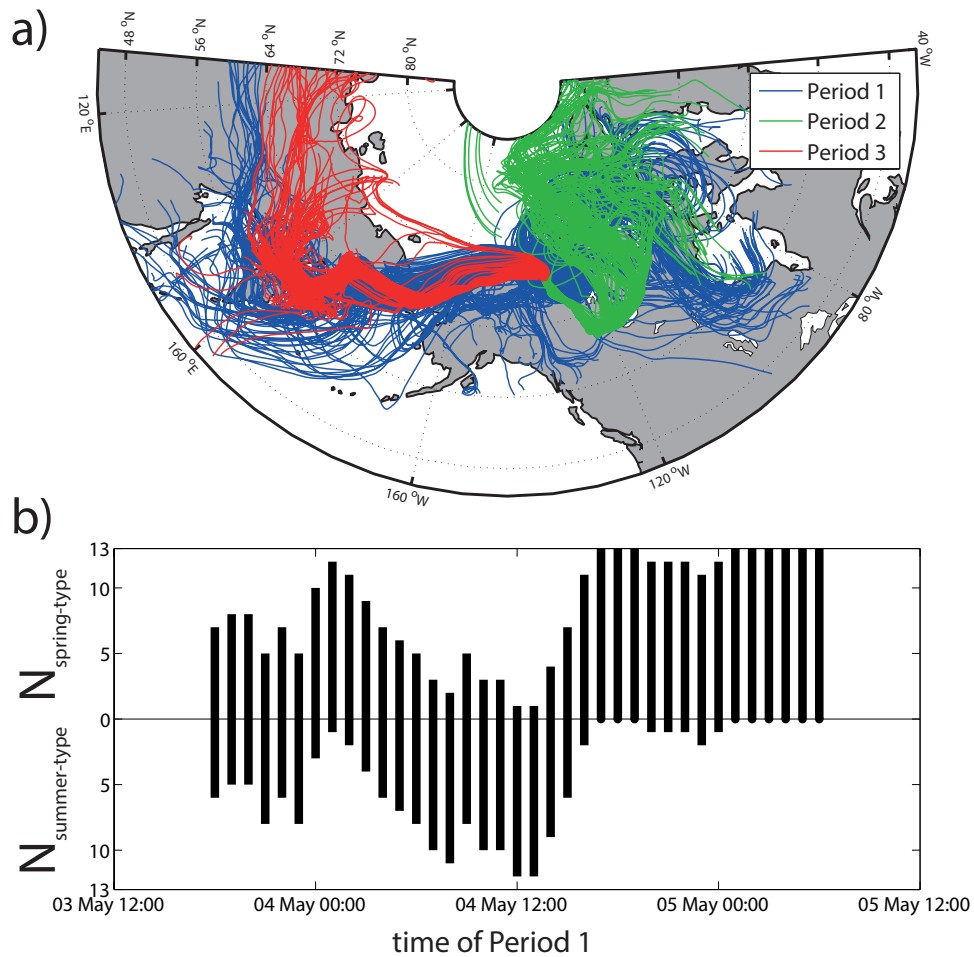

**Figure 6.** a) 10 day back trajectories of the three periods started in and around Tuktoyaktuk at a pressure level of $25\,\mathrm{hPa}$ above the surface pressure. b) The total number of 13 trajectories per hour of Period 1 was split up into the number of trajectories that came from east ($N_{spring-type}$) or west ($N_{summer-type}$) to illustrate the alternation of the air mass origin.

### 3.3 $PNSD$ of the three periods

Figure 7a shows the median of the $PNSD_c$ of the three periods discussed in Section 3.2, together with the 25 % and 75 % percentiles. They were computed to examine their dependence on the origin of the air mass. The $PNSD_c$ of Period 1 ($PNSD_{c1}$) and Period 3 ($PNSD_{c3}$) are bi-modal with an Aitken mode below $100\,\mathrm{nm}$ and an accumulation mode above $100\,\mathrm{nm}$ whereas that of Period 2 ($PNSD_{c2}$) only shows the accumulation mode. The large variability we observed in the shape of the $PNSD$ is typical for the transition period from Arctic spring to summer. $PNSD_{c2}$ is similar to the $PNSD$ that Tunved et al. (2013) observed for March and April on Svalbard. A direct comparison of $PNSD_{c2}$ and the median April $PNSD$ of Tunved et al. (2013) is shown in Figure 7b. The mono-modal accumulation mode aerosol is characteristic for the Arctic haze which mainly



consists of particulate organic matter (POM) and sulfate (Quinn et al., 2002). Single particle analysis of aerosol particles samples taken at the Zeppelin Station, Svalbard, that occurred before the transition to the Arctic summer showed a dominance of spherical organic like particles in the submicrometer range with an Eurasian influence (Behrenfeld et al., 2008). These Arctic haze aerosol particles typically are well aged (Heintzenberg, 1980; Quinn et al., 2002). Due to the shape of $PNSD_{c2}$, and since

the air mass of Period 2 has its origin in a region where conditions in May are still winterly, it is very likely that we observed a typical Arctic haze air mass. In contrast, $PNSD_{c3}$ is comparable to $PNSD$s that are reported by Tunved et al. (2013) for June and July. $PNSD_{c3}$ and the median June $PNSD$ of Tunved et al. (2013) are depicted in Figure 7b. In addition to the accumulation mode the bi-modal summer time Arctic $PNSD$ shows an Aitken mode which most likely originates from particles formed by new particle formation (Engvall et al., 2008; Wiedensohler et al., 2011). A common precursor gas for new particle

formation is dimethylsulfide (DMS) emitted from oceanic phytoplankton. This precursor is known to be more abundant during the Arctic summer when the marine biological activity has its maximum. An indicator for the presence of DMS is its oxidation product methanesulfonic acid (MSA) (Quinn et al., 2007). MSA also could be directly detected as component of the particulate matter itself in remote marine background aerosol and in plankton bloom areas (Zorn et al., 2008). Quinn et al. (2007) report the concentration of MSA for several Arctic measurement stations (e.g. Barrow and Alert - Tuktoyaktuk is located between the

two) during at least 7 years. The MSA concentration starts to increase in April and has two maxima during the summer time, where both maxima were observed in Alert as well as in Barrow (Quinn et al., 2007). The later maximum occurs in July and August and is due to the local productivity of phytoplankton while the surface water is free of ice. The earlier maximum that occurs around the time of our measurements, can be associated with long-range transport from marine source regions from the North Pacific (Li et al., 1993). This fits well to the source region we found for the air mass of $PNSD_{c3}$ and can explain

the presence of the Aitken mode particles. The minimum between the Aitken and the accumulation mode can be explained by previous cloud processes during which further material was added to activated droplets via aqueous phase oxidation. After the evaporation of cloud droplets this process creates the bi-modal $PNSD$ with the Hoppel-minimum (Hoppel et al., 1994). In our case the Hoppel-minimum can be found at around $90 \, \mathrm{nm}$. In Section 3.2 it is described that the back trajectories for time Period 1 altered between the two source regions. This observation is supported by the shape of $PNSD_{c1}$ which sug-

gests that both source regions contribute to the aerosol particles observed during this period. $PNSD_{c1}$ is bi-modal (similar to $PNSD_{c3}$) but with a less pronounced Hoppel-minimum and a distinct accumulation mode (similar to $PNSD_{c2}$). Due to the strongly alternating air mass origins during Period 1, the attempt to separate the two cases in $PNSD_1$ did not succeed, and the aerosol particle population reported for Period 1 in this study comprises a mixture between spring- and summer-type air masses. However, due to the absence of detailed information on the composition of the aerosol particles such considerations

remain speculative.





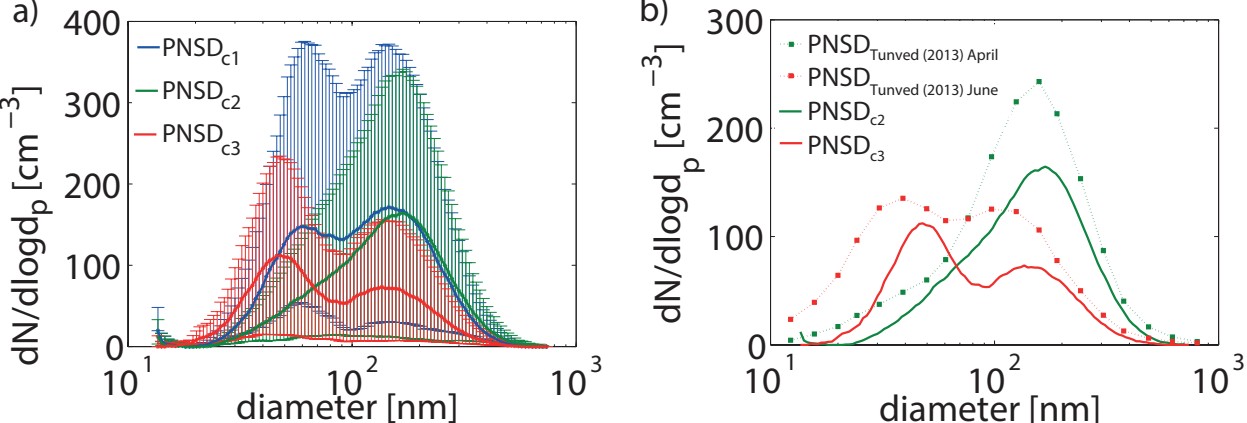

**Figure 7.** a) Thick lines show the median $PNSD$ of Period 1 (blue), Period 2 (green) and Period 3 (red). The thin vertical lines (same color code) indicate the range between the 25 % and 75 % percentiles. b) Comparison of the median $PNSD$ of Period 2 and 3 with 10 year median $PNSD$ of April and June observed by Tunved et al. (2013) on Svalbard. The $PNSD$s observed by Tunved et al. (2013) for April and June are comparable in shape with $PNSD_{c2}$ and $PNSD_{c3}$, respectively.

### 3.4 Critical diameter $d_{crit}$ and hygroscopicity parameter $\kappa$

In Section 2.2 it is described how the critical diameter $d_{crit}$ and the hygroscopicity parameter $\kappa$ can be determined based on the measured $N_{CCN}$ and $PNSD$. $d_{crit}$ and $\kappa$ were derived for (i) the whole measurement period using unfiltered $N_{CCN,0.1}$ and corresponding $PNSD$ and (ii) the three selected periods described above, using filtered $N_{CCN}$ at all $SS$ and $PNSD_c$. Figure

8 shows the time series of $N_{CN}$ and $N_{CCN}$ (mean concentration with standard deviation as error bars) in the upper panel, $d_{crit}$ in the middle panel and $\kappa$ in the lower panel. Note, data concerning all $SS$ are only available during the three selected periods that show no local pollution. In Section 3.1 we describe that the pollution occurred in the size range below $150\,\mathrm{nm}$. $N_{CCN}$ measured at $SS$ higher than $0.1\,\%$ are affected by local pollution due to the lower $d_{crit}$ ($d_{crit}$ is discussed in detail below) and thus are not analyzed for the whole measurement period except the three periods. The uncertainties for $d_{crit}$ and $\kappa$ as given

by the error bars were determined by the use of a Monte Carlo simulation. A detailed description of this method is given in Appendix A2. (The uncertainties for $d_{crit}$ are typically smaller than the symbol size.)

During the whole measurement period the unfiltered $N_{CCN,0.1}$ covers a range between less than 10 and $200\,cm^{-3}$ with a median of $45.2\,cm^{-3}$. The median $d_{crit}$ values as well as the median concentrations for the filtered $N_{CCN}$ are presented in 1. The corresponding inferred values for $\kappa$ are representative for aerosol particles with sizes in the size range close to $d_{crit}$ and

therefore can be assigned to the modes in the $PNSD$. Therefore, $\kappa$ derived for $SS = 0.1\,\%$ displays the hygroscopicity of the accumulation mode as $d_{crit}$ for $SS = 0.1\,\%$ is in the neighborhood of the maximum of this mode. $\kappa$ values for $SS = 0.2\,\%$ and $SS = 0.3\,\%$ are representative for the size range close to the Hoppel-minimum whereas $\kappa$ values for $SS = 0.5\,\%$ and $SS = 0.7\,\%$ are representative for the Aitken mode. The median $\kappa$ values for $SS = 0.1, 0.2, 0.3, 0.5$ and $0.7\,\%$ are $0.18, 0.28, 0.21, 0.23$ and $0.26$, respectively. These values are summarized in Table 1. The $\kappa$ values averaged over all $SS$ are $0.22, 0.23$



and 0.26 for time Period 1,2 and 3, respectively. We will discuss in the following how these $\kappa$ values relate to the measurement uncertainty.

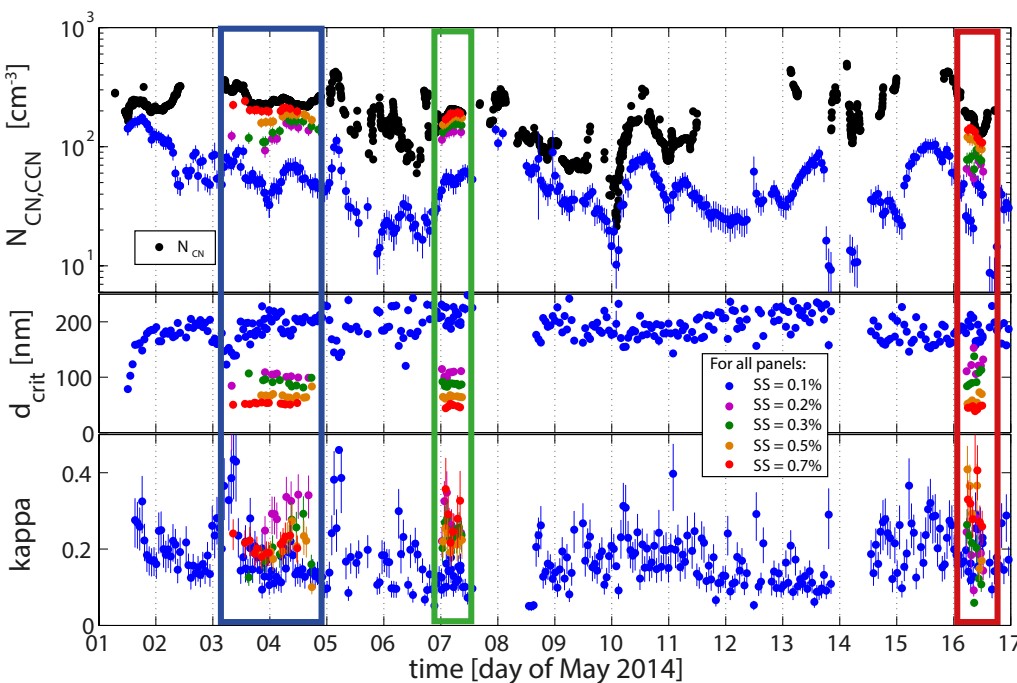

**Figure 8.** The time series of $N_{CN}$ and $N_{CCN}$ in the upper panel, $d_{crit}$ in the middle panel and the inferred $\kappa$ values in the lower panel. The error bars of $N_{CN}$ and $N_{CCN}$ show the standard deviation whereas the error bars of $d_{crit}$ and $\kappa$ show uncertainties that are determined by means of a Monte-Carlo simulation. The blue data points show the unfiltered data corresponding to $SS = 0.1\,\%$. The other colors correspond to the filtered data ($SS \geq 0.1\,\%$) of Period 1, 2 and 3, marked with the blue, green and red box, respectively.

**Table 1.** Median values of $N_{CCN}$, $d_{crit}$ and $\kappa$ for different $SS$. Values for $SS = 0.1\,\%$ are calculated using the unfiltered data that cover the entire measurement period, whereas the values for $SS = 0.2, 0.3, 0.5$ and $0.7\,\%$ are calculated using the filtered data of Period 1, 2 and 3.

| SS [%] | $N_{CCN}$ [cm$^{-3}$] | $d_{crit}$ [nm] | $\kappa$ |
|--------|------------------------|------------------|----------|
| 0.1 | 45 | 191 | 0.18 |
| 0.2 | 118 | 107 | 0.28 |
| 0.3 | 139 | 89 | 0.21 |
| 0.5 | 164 | 64 | 0.23 |
| 0.7 | 197 | 50 | 0.26 |





Figure 9a shows the probability density function (pdf) of all $\kappa$ values that are presented in the lower panel of Figure 8. The blue line displays the median of the inferred $\kappa$ values which is 0.23 and the red lines are the 5 and 95 % percentiles. The width between these percentiles, $\sigma_{s,allSS}$, amounts to 0.23. To check whether these inferred $\kappa$ values allow a physical interpretation of the variability of $\kappa$, the Monte Carlo simulation was used, as described in the Appendix A2. In short, using Monte Carlo

simulations, the uncertainty for each $\kappa$ value was determined separately based on uncertainties in $d_{crit}$, $N_{CCN}$ and $S_{crit}$. This uncertainty was again expressed as the width between the 5 and 95 % percentiles. The separate widths were averaged, yielding $\sigma_{MC,allSS}$, which was determined to be 0.16. To resolve physically driven changes, $\sigma_{s,allSS}$ should be significantly larger than $\sigma_{MC,allSS}$ (at least twice as large). But $\sigma_{s,allSS}/\sigma_{MC,allSS}$ only amounts to 1.44, which indicates that 70 % (= 1/1.44) of the variability in the observed $\kappa$ values is related to measurement uncertainties of the $PNSD$ and the $SS$ in the CCNc. For

corroboration, the same analysis was done based on a sub-set of all data. In Figure 9b the $\kappa$ values at $SS = 0.1\%$ are displayed as a probability density function with a median of 0.19 and a width between the 5 and 95 % ($\sigma_{s,0.1}$) percentiles of 0.23. The width between the 5 and 95 % percentiles of the Monte Carlo simulation ($\sigma_{MC,0.1}$, only for $SS = 0.1\%$) is 0.14 so that the ratio between both is 1.64. Hence 60 % of the variability in the observed $\kappa$ values at $SS = 0.1\%$ is related to measurement uncertainties. Summarizing, it can be stated that our observed $\kappa$ values show no significant dependencies on the $SS$ or the air

mass origin that can be resolved with our set up and method.

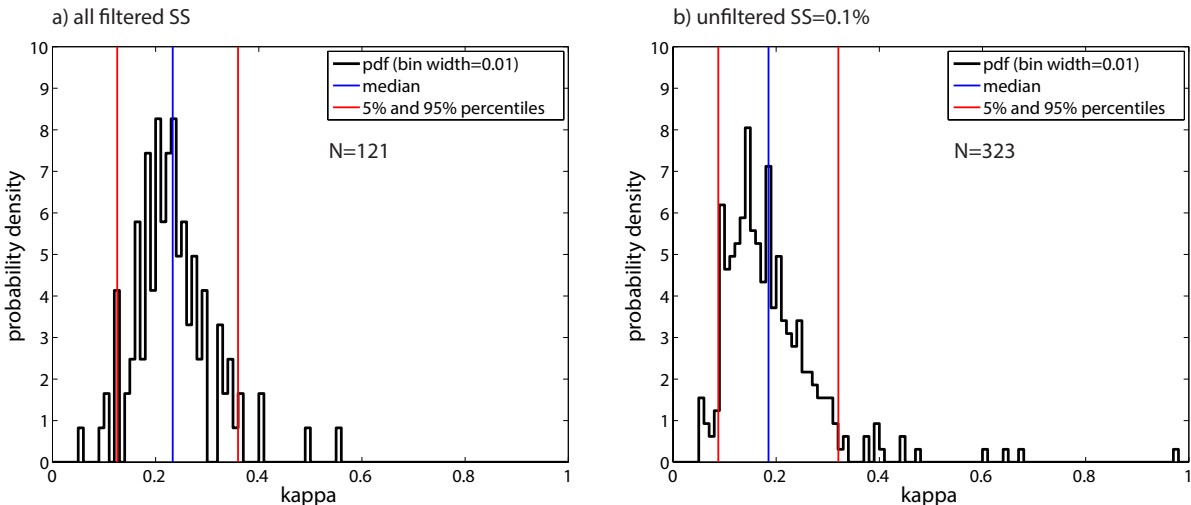

**Figure 9.** a) Probability density function of all inferred $\kappa$ values (filtered values of all supersaturations) of the lower panel of Figure 8. b) Probability density function for $\kappa$ values at $SS = 0.1\%$ (inferred from the unfiltered CCN and $PNSD$ data of Figure 4). The blue line displays the median and the red lines show the 5 and 95 % percentiles of the probability density function.

Kammermann et al. (2010) measured $N_{CCN}$ and inferred $d_{crit}$ and $\kappa$ in a subpolar environment during a ground based measurement campaign in north Sweden for $SS$ from 0.1 to 0.7 % in July 2007. They report $\kappa$ values in the range of 0.07 to 0.21. Moore et al. (2011) and Lathem et al. (2013) report $\kappa$ values from airborne measurements in Alaska (April 2008) and North



Canada (June to July 2008), respectively. Both observed values between 0.05 and 0.3 within a $SS$ range of 0.1 to 0.6 %. Lower $\kappa$ are likely to indicate a higher organic fraction in these environments, and particularly for Kammermann et al. (2010), the lowest values can be explained by the local proximity to the Stordalen mire which is known to emit biogenic precursors of organic aerosol particles. In a modeling study by Pringle et al. (2010), the annual mean $\kappa$ values at the surface in the region around Tuktoyaktuk were approximately 0.3. Overall the $\kappa$ values derived in this study are comparable to previously published values.

### 3.5 Comparison of height resolved airborne and ground based $PNSD$s

During the campaign 4 overflights of the Polar 6 research aircraft were performed. Overflight 1 and 2 were single overflights at a constant altitude of 600 m and 200 m, respectively. During Overflight 3 and 4 eight legs in altitudes between 300 m and 3000 m were flown. The comparison of the airborne and ground based measured $PNSD$s of the four overflights is shown in Figure 10. Arrows in Figure 4 indicate the times when the four overflights took place. For Overflight 1 and 2 simultaneous measurements of filtered $PNSD_c$ exist. For Overflight 3 and 4 the closest filtered $PNSD_c$ measurements have a temporal distance of 7 and 6 hours to the time of overflight, respectively. Hence, for the comparison in case of Overflight 3 and 4 the unfiltered $PNSD_d$ measurements are used. The airborne $PNSD$s measured by means of an UHSAS were recorded with a time resolution of 1 s and extrapolated to standard pressure (1013.25 hPa). In Figure 10 the UHSAS distributions are generally displayed as the median of 100 measured distributions. Additional bars that indicate the range between the 25 and 75 % percentiles are added to the distributions of Overflight 1 and 2. $PNSD_c$ and $PNSD_d$, which were measured at the ground, are shown for ambient pressure.

For Overflight 1 and 2, the ground based $PNSD_c$s agree well with the airborne $PNSD$ in the overlapping size range of 70 nm to 736.5 nm, where airborne measurements were done at 600 m and 200 m, respectively. Vertical temperature profiles observed by radiosondes over Inuvik show temperature inversions at altitudes of 1500 m and 700 m for Overflight 1 and 2, respectively (not shown here). This indicates that during these two distinct time periods the ground based measurements of $PNSD$s are representative of the atmospheric boundary layer.

For Overflights 3 and 4, the measured $PNSD$s varied with respect to the particle number concentration and shape for the flights in different altitudes between 300 m and 3000 m. The airborne $PNSD$s of Overflight 3 show the same shape at all eight heights with a clear decrease of the number concentration at lower heights. The integration of the $PNSD$ measured at 300 m (black line) gives a total particle number concentration of 24 particles per $cm^3$, whereas it is $512\,cm^{-3}$ at an altitude of 3000 m, i.e., twenty times higher. However the shape of the $PNSD$s does not change with height as all distributions are mono-modal with a maximum at approximately 150 nm. The ground based $PNSD_d$ in the size range above 150 nm agrees best with the airborne $PNSD$ that was measured at 1200 m. At smaller sizes no comparison can be done, as the local pollution produces a large mode below 150 nm. The ambient temperature recorded at the Polar 6 aircraft during Overflight 3 indicates a temperature inversion near an altitude of 2000 m. For further investigation back trajectories at altitudes of 1000, 2000 and 3000 m were calculated to investigate the history of air masses at different altitudes. The trajectories arriving at altitudes of 1000 and 2000 m show an air mass origin in the area of the Northern Pacific, comparable to time Period 3 in Section 3.2. But



the trajectory that arrived in 3000 m indicates an air mass origin in the central Arctic and over Greenland. Hence, the origin of the air masses and the relatively higher particle number concentration in the accumulation mode of the $PNSD$ may indicate that the typical aged Arctic spring aerosol, that was observed during Period 2, is present above the temperature inversion. This aerosol could be mixed down to lower layers accompanied by a dilution process, however, aerosol observed at the lower

levels is likely mostly of a different origin. Overflight 4 shows that the airborne $PNSD$s also may differ in shape depending on the height. The $PNSD$s between 1750 and 3000 m are mono-modal with a maximum between 100 and 200 nm. Also a comparably high particle number concentration was measured at an altitude of 3000 m. The $PNSD$s at lower heights imply a second mode below 100 nm. This Aitken mode is also present in the ground based $PNSD_d$ which fits the airborne $PNSD$s that were measured below 1200 m. The air masses above 1750 m show characteristics of the typical aged Arctic spring-type

aerosol (comparable to Period 2) whereas the air masses below 1200 m seem to consist of aerosol of marine origin (comparable to Period 3). For Overflight 4, two temperature inversions were recorded between 2500 and 3000 m. The temperature inversions and the different shapes of the $PNSD$s are indicative for the presence of different air masses during Overflight 4, although air mass back trajectories that arrived at 1000, 2000 and 3000 m indicate an air mass origin over the Northern Pacific for all three heights. Stone et al. (2014) explain that layering of Arctic aerosol, as we observed it during Overflight 4, is a function of

where the aerosol particle sources are located. Thereby a crucial factor are the different pathways of aerosol transport in the lower Arctic troposphere. The cold air of the lower Arctic troposphere is covered by surfaces of constant potential temperature and forms a dome over the Arctic (Law and Stohl, 2007). According to Stohl (2006) three transport pathways are possible: 1) low-level transport followed by ascent along the surfaces of constant potential temperature; 2) only low-level transport; 3) uplift outside the Arctic followed by transport in the upper troposphere and descent in the Arctic. It is likely that the aerosol

particles we observed in the upper levels of Overflights 3 and 4 were transported via pathway 1 or 3 whereas pathway 2 might be responsible for the occurrence of the bi-modal $PNSD$ below 1200 m during Overflight 4.

Note that the altitude resolved $PNSD$s presented here only represent a short snapshot in time. Hence, our observations do not describe how the transition from Arctic spring to summer affects the Arctic $PNSD$ in the different lower layers of the troposphere. However, the measurements during all four overflights show that the ground based $PNSD$ is similar

to the airborne $PNSD$ of the lowest tropospheric layers. Therefore it can be excluded that local natural sources contribute significantly to our measurements during the observed time period at least after removing signals from local pollution. It is more likely that aerosol is advected via long range transport from lower latitudes in different height layers and mixed down in the lower Arctic troposphere.



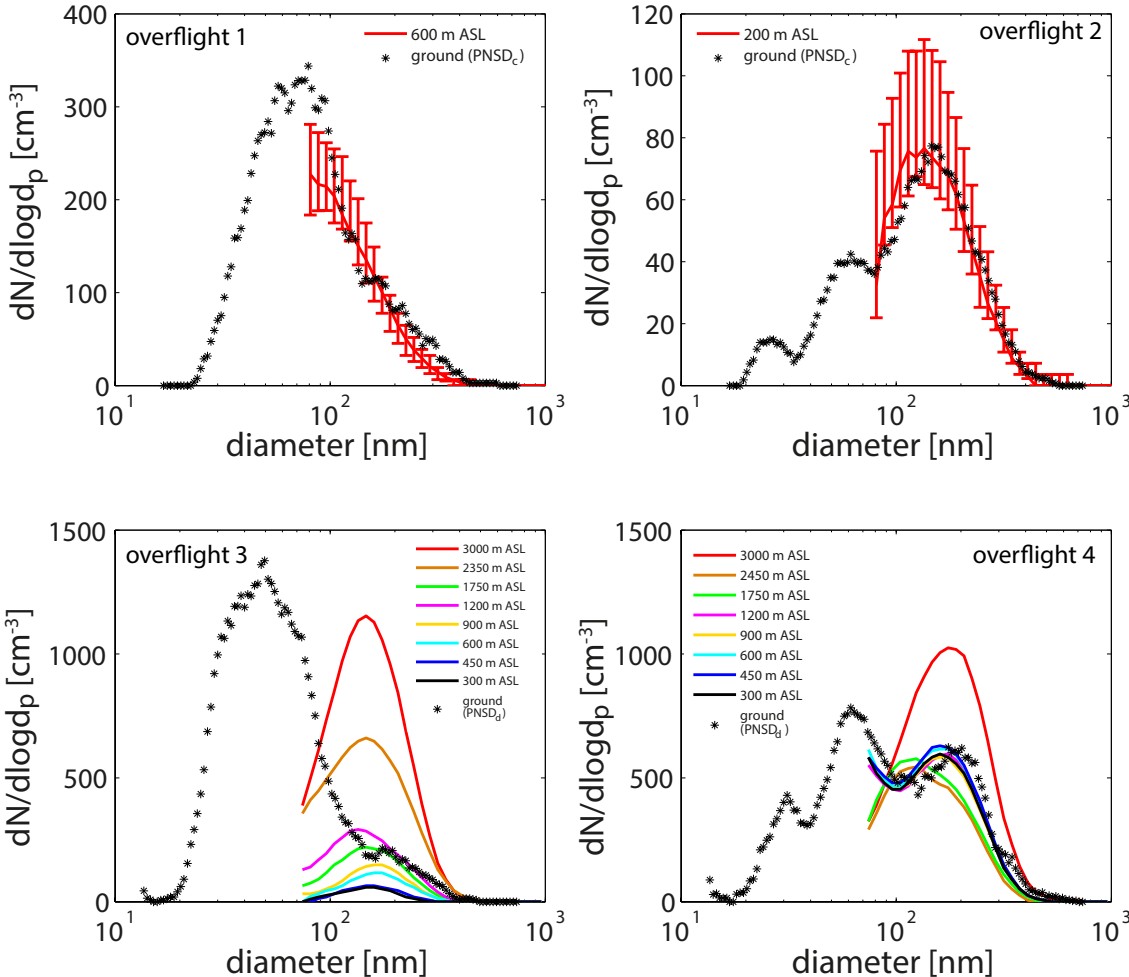

**Figure 10.** $PNSD$s measured at the ground based station and on the Polar 6 research aircraft during four overflights. Overflight 1 and 2 took place in $200\,m$ and $600\,m$, whereas Overflight 3 and 4 were profile flights at altitudes between 300 and 3000 m. Airborne measurements from $70\,nm$ up to $1\,\mu m$ were done using an UHSAS, while ground based $PNSD_c$s and $PNSD_d$s of aerosol particle diameters from $13.6\,nm$ up to $736.5\,nm$ were measured using a SMPS as indicated in the setup of Figure 1.

## 4   Summary and conclusion

Arctic CCN and aerosol particle properties were measured during the RACEPAC campaign in May 2014. Due to the occurrence of local pollution, a filtering procedure had to be applied before the data set was further evaluated to obtain estimates for the background conditions. The local pollution caused peaks in up to more than $10000\,cm^{-3}$ with a typical temporal duration





of 1 to 5 minutes in $N_{CN}$. A comparison of the $PNSD$s of the polluted and the unpolluted periods shows that the local pollution significantly contributed to the measured particle number concentration below a size of 150 nm. As a consequence of this, $N_{CCN}$ and $\kappa$ obtained for $SS = 0.1\%$ could be analyzed for the whole measurement period, as these were not affected by pollution events of particle sizes below 150 nm, while $N_{CCN}$ and $\kappa$ at all other measured $SS$ as well as the $PNSD$s were analyzed for three distinct time periods only. 10 day back trajectories that were computed for the three periods showed that air masses from two different origins were investigated. Air masses with an origin in North-East Canada were typically dominated by Arctic haze. The corresponding mono-modal $PNSD$ shows an accumulation mode which most likely contains well aged particles that have an Eurasian origin (Behrenfeld et al., 2008). The other origin is the region of the North Pacific and Eastern Russia. The corresponding bi-modal $PNSD$ shows an additional mode of smaller particles that may be attributed to new particle formation and growth potentially due to oxidation products of marine emissions of DMS (Engvall et al., 2008; Wiedensohler et al., 2011). This variability in the $PNSD$ is typical for the transition from Arctic spring to summer during April, May and June (Engvall et al., 2008). CCN number concentrations were found to cover a range between less than 10 and 250 $cm^{-3}$ for $SS$ between 0.1 and 0.7 %, respectively. Applying the $\kappa$-Köhler theory (Petters and Kreidenweis, 2007) the hygroscopicity parameter $\kappa$ was inferred. The median $\kappa$ of all $SS$ and all three periods is 0.23. At $SS = 0.1\%$, for which the whole measurement period could be evaluated, we found a mean $\kappa$ of 0.19 . The estimated random errors typically exceed the observed variation in the observed $\kappa$ values. Consequently, it was not possible to distinguish $\kappa$ values related to different air masses or particle sizes.

Simultaneous measurements at the ground based measuring station in Tuktoyaktuk and on the research aircraft Polar 6 show a qualitative good agreement of ground based $PNSD$s with $PNSD$s of the lowest tropospheric layers (up to 1200 m when measurements at this height were present) during four overflights. Hence, it can be excluded that local natural sources contribute significantly to our ground based measurements during the observed time period and that the ground based measurements of $PNSD_c$s (without influence of local pollution) are representative for the atmospheric boundary layer in the area of Tuktoyaktuk during the measurement period. Moreover two profile flights show that the $PNSD$s measured inside and above the atmospheric boundary layer can vary in shape and integrated particle number concentration. We observed the largest particle number concentrations in the highest layer (3000 m). It can be assumed that the aerosol is advected via long range transport from lower latitudes in different height layers and mixed down in the lower Arctic troposphere.

Certainly, the underlying data base that was used for this analysis is small. However as the conditions in the Arctic are changing very rapidly such measurements do have a value for future reference as they document the situation at a specific time period during the change.

## Appendix A: Validation of the instruments and corrections that have been done

### A1 Post-processing and filtering of the raw data set

Measurements of $N_{CN}$, $N_{CCN}$ and $PNSD$ were contaminated due to local particle sources, so that a filter routine had to be applied. The filter completely removes data points at time periods during which the pollution occurred. $N_{CN}$ is the parameter




that is most sensitive to the pollution since it was detected with the lowest time resolution (1 s) and pollution occurred in a size range smaller than 150 nm in diameter, which is covered by the CPC-3010. Pollution events were identified due to a fast (some seconds) and intensive increase of $N_{CN}$, which is well visible in the $N_{CN}$ time series. Consequently, the gradient in the $N_{CN}$ time series was used as a filter criterion. The peaks that occurred due to local pollution events could be identified best by

searching for an absolute gradient between two $N_{CN}$ measuring points of at least $\pm$ 20 particles per $cm^3 s$. For further analysis, $N_{CCN}$ and $PNSD$ that were measured in a time span of 400 s before and after a pollution peak occurred were neglected. The 400 s originated from the sampling frequency of $N_{CCN}$ (400 s) and $PNSD$ (318 s).

During the measurement period technical problems occurred with the CPC-3025 which was a part of the SMPS system. This resulted in a not uniform consistency between $N_{CN}$ measured with the CPC-3010 and the $N_{CN}$ of the integrated $PNSD$

measured with a CPC-3025. Hence, the $PNSD$s were variably corrected so that the integrated total number concentration was consistent with $N_{CN}$ measured with the CPC-3010.

## A2   Determination and error estimation of $d_{crit}$ and $\kappa$ using a Monte Carlo simulation (MCS)

Measurements of $PNSD$ and $N_{CCN}$ come along with device specific uncertainties. For instance the particle diameter that is selected with a DMA can be assumed to have an uncertainty of 3 % and the measured particle number concentration an

uncertainty of 5 % corresponding to 1 standard deviation, respectively (Gysel and Stratmann, 2013). Moreover, the effective $SS$ in the CCN counter has a relative uncertainty of 3.5 % for $SS$ above 0.2 % corresponding to 1 standard deviation. These uncertainties have been inferred from several $SS$ calibrations that were performed at the Leibniz Institute for Tropospheric Research (TROPOS). Below $SS = 0.2$ % the same absolute uncertainty as for $SS = 0.2$ % can be assumed (Gysel and Stratmann, 2013). To consider the impact of these uncertainties on $d_{crit}$ and $\kappa$ in a realistic way, a Monte Carlo simulation (MCS) based

on random normal distributions was used. This following general equation was applied:

$$s_{MC} = s + (s * u * p),  \tag{A1}$$

where $u$ is the relative uncertainty, $p$ a random number, $s$ is the measured signal and $s_{MC}$ the resulting MCS signal. This was done for 10000 random and normally distributed numbers $p$, with a mean of 0 and a standard deviation of 1, which then results in 10000 values for $s_{MC}$ with a variability that is characterized by $u$.

In a first step, the uncertainty in $d_{crit}$ was obtained by a MCS based on one exemplary $PNSD$, the related $N_{CCN}$ and a 5 % uncertainty in the particle number concentration. Equation A1 was used to vary the particle number concentration of each size bin of the $PNSD$ to calculate 10000 $d_{crit}$ values, of which a distribution is shown in Figure A1a. The mean and standard deviation of these 10000 $d_{crit}$ values can be taken from this distribution, and the overall uncertainty in $d_{crit}$ was derived from those values together with the 3 % uncertainty in the particle sizing due to the DMA, using error propagation. This was then

done for all $PNSD$s. The resulting uncertainties are shown as error bars in the middle panel of Figure 8.
$\kappa$ and the corresponding error bars in the lower panel of Figure 8 are inferred by means of Eq. 1 where $d_{crit}$ and $S_{crit}$, which is the effective $SS$ of the CCN counter, are 10000 times Monte Carlo simulated (same procedure as for $d_{crit}$). Since the



connection between $\kappa$ and $SS$ is logarithmic the resulting distribution of the 10000 $\kappa$ values is a log-normal distribution, as can be seen in Figure A1b exemplarily for one case. Consequently, our final inferred $\kappa$ and its uncertainty are the median and the 5 and 95 % percentiles of this distribution, respectively. The average of all widths between the 5 and 95 % percentiles is the value we compared with the width between the 5 and 95 % percentiles of all median $\kappa$ values to make a statement about the

significance of our results.

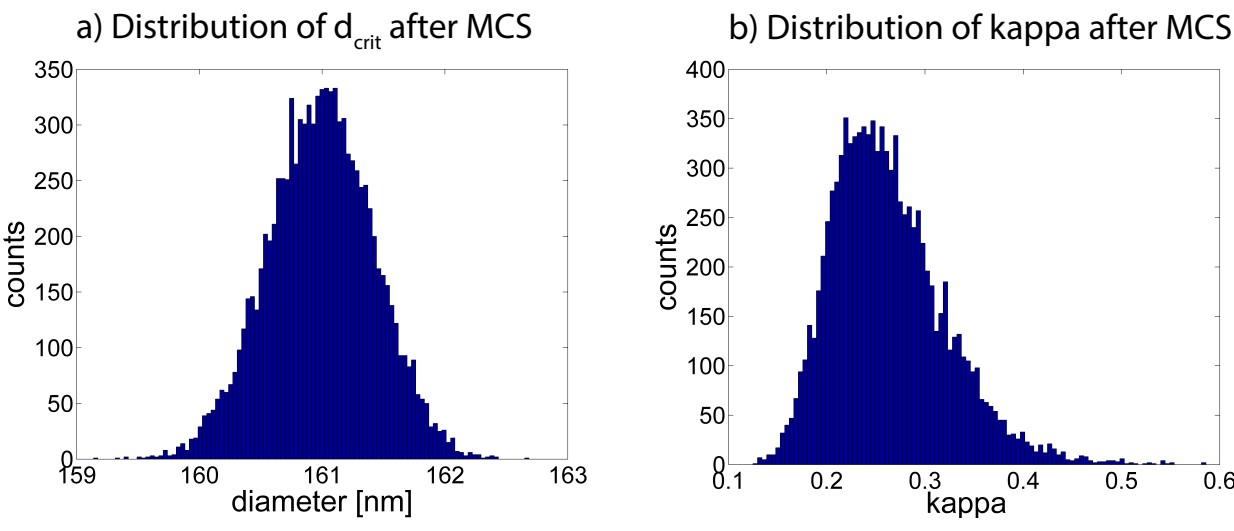

**Figure A1.** a) Distribution of 10000 $d_{crit}$ values after applying the MCS. The mean and the standard deviation of this distribution are the final $d_{crit}$ and its uncertainty due to the 5 % uncertainty in the particle number concentration of each size bin in the $PNSD$, respectively. b) Distribution of 10000 $\kappa$ values after applying the MCS. As this results in a log-normal distribution of $\kappa$ values it is more appropriate to use the median and percentiles as the final $\kappa$ value and its uncertainty, respectively.

*Acknowledgements.* This work was part of RACEPAC (Radiation-Aerosol-Cloud Experiment in the Arctic Circle) and we thank the RACEPAC coordination team (University of Leipzig, TROPOS Leipzig, MPI-C Mainz) for initiating and organizing the campaign. Furthermore, we acknowledge the AWI (Alfred-Wegener-Institut, Helmholtz-Zentrum für Polar und Meeresforschung) Bremerhaven for fruitful scientific collaboration and providing the aircraft. We also thank the Canadian Polar Continental Shelf Program (PCSP), the Aurora Research Institute

and their staff for logistic and administrative support, in particular for obtaining the necessary licences (PCSP Project Number 71614, and ARI Research Licence Number 15407), the permissions to operate the station, and to conduct the overflights. We gratefully acknowledge the support and help from the people of the Inuvialuit Hamlet of Tuktoyaktuk. Significant financial support was also provided by internal sources of the Max-Planck-Society for the setup and operation of the Tuktoyaktuk station, as well as for the execution of the Polar 6 resaearch flights. Further this work was partially funded by the EU FP7-ENV-2013 program "Impact of Biogenic vs. Anthropogenic emissions on Clouds and

Climate: towards a Holistic UnderStanding" (BACCHUS), project number 603445. T.B. Kristensen acknowledges funding from the German Federal Ministry of Education and Research (BMBF) project no. 01LK1222B.



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
