# Peer review of "Measurements of aerosol and CCN properties in the Mackenzie River delta (Canadian Arctic) during Spring-Summer transition in May 2014"

_Atmospheric Chemistry and Physics, 2017_

## Referee Comment (RC1) · Anonymous Referee #2 · 21 Sep 2017

General Comments:

The manuscript by Herenz et al discusses an interpretation of data collected at a ground site in Tuktoyaktuk, Canada, in association with aircraft data during May 2014. This time period is a 'shoulder season' in the Arctic, where the spring 'Arctic Haze' influence is waning and the atmosphere is beginning to take on summer-like characteristics. The paper discusses two types of air mass influences observed at Tuktoyaktuk, which the authors have called 'spring-type' and 'summer-type'. The spring-type air has characteristics similar to Arctic Haze and the summer-type air has an added influence from Aitken mode particles. The measurements and their analysis are technically

sound, and data filtering has been presented appropriately. Some of the conclusions and interpretations of the data are confusing or incomplete in the opinion of this reviewer, and have been pointed out below. In addition, the labeling of 'spring-type' and 'summer-type' is perhaps generous (particularly 'summer-type') as it is not shown that the source region or processes shaping the size distribution during the study period are similar to those in the heart of the summer season (July-August). It is perhaps prudent to label the Aitken-mode containing PNSDs as simply 'N. Pacific', since the study does well to describe the highly time-resolved oscillations in air masses during near the spring-summer transition, and correlates this with one of two seasonal maxima in methanesulfonic acid (MSA) (Leaitch et al., 2013; Quinn et al., 2007). As a curious point, the authors performed this research in the Canadian Arctic, yet have not cited much of the existing research that has been conducted in the region, so the suggested or argument-supporting citations provided in this review have been skewed slightly to correct for this fact.

This paper could be accepted to Atmospheric Chemistry and Physics pending major revisions to the discussion and interpretation of the data.

Specific comments:

The abstract of the paper highlights a 'rapid transition from Arctic spring to summer that took place during the measurement period.' This claim is difficult to support directly with the extent of analysis that took presented in this study. Such a claim would be more amenable to inter-seasonal data that are analyzed at high temporal resolution – such a study does not exist to the knowledge of this reviewer – but the fact that two types of air masses influence this site is certainly supported within the manuscript. One longer period of focused analysis in the present manuscript indicates an oscillation between the so-called 'spring' and 'summer' influences on the scale of hours-to-days (Period 1, Figure 6b). Characterizing this oscillating influence as a rapid transition is perhaps technically accurate in some sense; it is also misleading, as the abstract leads the reader to believe that the site will change singularly from 'spring' to 'summer'

like a step function. The presented data suggests this is not occurring, and rather the measurement site is alternately influenced by two different source regions (one of which is debatably 'summer-like'), so it is recommended that the abstract be rephrased. The introduction of the paper is quite well written and clear, with some very minor exceptions noted in the detail comments below.

Overall, the measurements presented in the study were analyzed systematically, clearly, and with substantial attention to quantitative detail, including a comprehensive error analysis. Both the size distribution data and the CCN data were used to appropriate lengths, considering uncertainties. The authors should be commended for working with this challenging data set collected in a difficult measurement environment.

The authors invoke cloud processing as the source of the bi-modal character of the PNSD_C3 size distribution (Page 13, lines 20-24) and this is, indeed, one possibility; however, different explanations that are equally supported by the measurements (as presented) are possible. The interpretation of the factors driving the shape of PNSD_C3 has broader consequences within this study, so this is quite an important point. In addition to the cloud processing argument presented by the authors for the bi-modal character of PNSD_C3, possible explanations include:

1. The PNSD_C3 average size distribution may be a mixture of two types of particles: newly formed particles in the Aitken mode and aged particles in the accumulation mode. Evidence for a mechanism to produce such a scenario has been provided by detailed analysis of size distributions from remote locations in the Arctic (e.g., Collins et al., 2017) wherein new particles may be forming on different days where growth of the existing population occurs simultaneously, causing particles to grow into the accumulation mode size range.

2. The PNSD_C1 mode sizes are quite similar to those of PNSD_C3, and it has been posited that Period 1 has a mixed character of spring-like and summer-like influences. Could there be additional source variability within Period 3 that gives rise to the bimodal character? Indeed on page 17-18, the authors discuss the vertical structure of air masses, suggesting that during Overflight 3, spring-like Arctic air resides above a surface inversion and could mix down into the surface layer. This mechanism would produce a size distribution with characteristics similar to those observed in PNSD_C3.

3. Recent studies at Utqiagvik, AK have shown the strong influence that fossil fuel extraction and processing facilities can have on the aerosol in this region (Gunsch et al., 2017; Kolesar et al., 2017) and could contribute to the shape of PNSD_C3 since the Period 3 trajectories pass over Northern Alaska.

If uncertainties in the CCN data were smaller, the kappa values of each mode may have provided useful in constraining these four options (one provided by the authors plus three above). Can the authors constrain the system further using any supplementary data or analysis that is not already provided?

The authors conclude in Section 3.5 that the shape of the PNSD is driven by transport and that local sources are not active in shaping the distribution. This is somewhat antithetic to the behaviour of canonical summertime Arctic aerosol which are thought to have a substantial contribution from new particle formation (Asmi et al., 2016; Croft et al., 2016; Tunved et al., 2013) based on sources within the polar dome via isentropic transport, which the authors discuss. Wet deposition processes (e.g., nucleation and precipitation scavenging) play a major role in shaping the size distribution in summer as well (Browse et al., 2012; Croft et al., 2016), which is not discussed or considered in the manuscript. Recent studies from the Canadian Arctic suggest that transport in heart of the summer is either from the higher Arctic latitudes or from the Canadian boreal forest (Collins et al., 2017; Mungall et al., 2016; Sharma et al., 2012; Wentworth et al., 2016), rather than from the North Pacific.

Critical question: Can the authors provide a stronger connection between the characteristics of the summer Arctic atmosphere and the Period 3-type air masses outside of the presence of the Aitken mode? Should the 'summer-type' (Period 3) air masses

be thought of as truly characteristic of summer, or are these specific to a transient seasonal transport pathway from the N. Pacific?

Page 3, lines 14-15: The authors claim a lack of in-situ measurements of CCN in the Arctic region. Recently, CCN and cloud property studies from the NETCARE campaigns have documented aspects of aerosol-cloud interactions, including airborne estimations of activation diameters and chemical effects on CCN activity, specifically in the Canadian Arctic (Burkart et al., 2017; Leaitch et al., 2016; Willis et al., 2016). Other studies, including those referenced in the manuscript, have studied CCN concentrations and properties in other regions (e.g., Moore et al., 2011).

Page 8, lines 11-13: Since this study was conducted in the Canadian Arctic, it is worthwhile to compare with seasonal aerosol concentrations at Alert, Canada and Utqiagvik (Barrow), AK. If the authors desired to make a more pan-Arctic comparison, inclusion of observations made at Station Nord, Greenland (Nguyen et al., 2016) and Tiksi, Russia (Asmi et al., 2016) should also be discussed.

Page 10, lines 9-10: It is difficult to see the size distribution results as a corroboration of the prior discussion of N_CNraw, since this is the same data shown in a different manner. On page 9, the N_CCN and N_CNraw>150nm were shown and noted as being correlated since the smaller particles are due to pollution. Figure 5 is simply a re-statement of the same information since the accumulation mode size is 150 nm. It is likely that the choice of 150 nm as a cutoff for the metric in Figure 4 was deliberate, and the data do certainly tell a convincing story about the difference between polluted and 'filtered' data, but these figures represent one piece of evidence, rather than two separate, corroborating pieces of evidence. If other types of techniques distinguished polluted from non-polluted data (chemical or physicochemical observations), then these would corroborate the number size distribution differences between polluted and filtered cases as presented.

Page 11, lines 19-20: When discussing the Period 3 air masses, no discussion on

[Figure]

similarity to prior studies is provided to support the notion that these air masses are indicative of what might be found in the actual summer months at Tuktoyaktuk. On page 13, lines 17-20, a statement about corroborating evidence for N. Pacific sources of air is made in reference to trends in MSA – but it is not shown that this type of air mass is truly typical of Arctic 'summer' as labeled.

Figure 6: It would be helpful to understand the behavior of the ensemble of back trajectories in the vertical dimension to some degree, so that impacts of surface sources along the trajectory could be understood. Is it possible to include some information on the vertical position of the trajectory or each ensemble of trajectories as a function of time or position?

Technical Corrections:

Page 1, line 4: "...were indicative for the rapid..." change to "...were indicative of the rapid..."

Page 8, line 11: change to "time periods where the measurements"

Page 9, lines 8-10: These two sentences are hard to follow. It is recommended that the sentences be re-phrased, especially when referring to "it is associated with lower sampling statistics", but the prior sentence refers to two different metrics.

References

Asmi, E., Kondratyev, V., Brus, D., Laurila, T., Lihavainen, H., Backman, J., Vakkari, V., Aurela, M., Hatakka, J., Viisanen, Y., Uttal, T., Ivakhov, V. and Makshtas, A.: Aerosol size distribution seasonal characteristics measured in Tiksi, Russian Arctic, Atmos. Chem. Phys., 16(3), 1271–1287, doi:10.5194/acp-16-1271-2016, 2016.

Browse, J., Carslaw, K. S., Arnold, S. R., Pringle, K. and Boucher, O.: The scavenging processes controlling the seasonal cycle in Arctic sulphate and black carbon aerosol, Atmos. Chem. Phys., 12, 6775–6798, doi:10.5194/acp-12-6775-2012, 2012.

Burkart, J., Willis, M. D., Bozem, H., Thomas, J. L., Law, K., Hoor, P., Aliabadi, A. A., Köllner, F., Schneider, J., Herber, A., Abbatt, J. P. D. and Leaitch, W. R.: Summertime observations of ultrafine particles and cloud condensation nuclei from the boundary layer to the free troposphere in the Arctic, Atmos. Chem. Phys., 17, 5515–5535, doi:10.5194/acp-17-5515-2017, 2017.

Collins, D. B., Burkart, J., Chang, R. Y.-W., Lizotte, M., Boivin-Rioux, A., Blais, M., Mungall, E. L., Boyer, M., Irish, V. E., Masse, G., Kunkel, D., Tremblay, J.-É., Papakyriakou, T., Bertram, A. K., Bozem, H., Gosselin, M., Levasseur, M. and Abbatt, J. P. D.: Frequent Ultrafine Particle Formation and Growth in the Canadian Arctic Marine Environment, Atmos. Chem. Phys. Discuss., 2017.

Croft, B., Martin, R. V., Leaitch, W. R., Tunved, P., Breider, T. J., D'Andrea, S. D. and Pierce, J. R.: Processes controlling the annual cycle of Arctic aerosol number and size distributions, Atmos. Chem. Phys., 16(6), 3665–3682, doi:10.5194/acp-16-3665-2016, 2016.

Gunsch, M. J., Kirpes, R. M., Kolesar, K. R., Barrett, T. E., China, S., Sheesley, R. J., Laskin, A., Wiedensohler, A., Tuch, T. and Pratt, K. A.: Contributions of transported Prudhoe Bay oil field emissions to the aerosol population in UtqiaĄ̈avik, Alaska, Atmos. Chem. Phys., 17, 10879–10892, doi:10.5194/acp-17-10879-2017, 2017.

Kolesar, K. R., Cellini, J., Peterson, P. K., Jefferson, A., Tuch, T., Birmili, W., Wiedensohler, A. and Pratt, K. A.: Effect of Prudhoe Bay emissions on atmospheric aerosol growth events observed in UtqiaĄ̈avik (Barrow), Alaska, Atmos. Environ., 152, 146–155, doi:10.1016/j.atmosenv.2016.12.019, 2017.

Leaitch, W. R., Sharma, S., Huang, L., Toom-Sauntry, D., Chivulescu, A., Macdonald, A. M., von Salzen, K., Pierce, J. R., Bertram, A. K., Schroder, J. C., Shantz, N. C., Chang, R. Y.-W. and Norman, A.-L.: Dimethyl sulfide control of the clean summertime Arctic aerosol and cloud, Elem. Sci. Anthr., 1(1), 17, doi:10.12952/journal.elementa.000017, 2013.

Leaitch, W. R., Korolev, A., Aliabadi, A. A., Burkart, J., Willis, M. D., Abbatt, J. P. D., Bozem, H., Hoor, P., Köllner, F., Schneider, J., Herber, A., Konrad, C. and Brauner, R.: Effects of 20–100 nm particles on liquid clouds in the clean summertime Arctic, Atmos. Chem. Phys, 16, 11107–11124, doi:10.5194/acp-16-11107-2016, 2016.

Moore, R. H., Bahreini, R., Brock, C. A., Froyd, K. D., Cozic, J., Holloway, J. S., Middlebrook, A. M., Murphy, D. M. and Nenes, A.: Hygroscopicity and composition of Alaskan Arctic CCN during April 2008, Atmos. Chem. Phys., 11(22), 11807–11825, doi:10.5194/acp-11-11807-2011, 2011.

Mungall, E. L., Croft, B., Lizotte, M., Thomas, J. L., Murphy, J. G., Levasseur, M., Martin, R. V., Wentzell, J. J. B., Liggio, J. and Abbatt, J. P. D.: Dimethyl sulfide in the summertime Arctic atmosphere: measurements and source sensitivity simulations, Atmos. Chem. Phys., 16(11), 6665–6680, doi:10.5194/acp-16-6665-2016, 2016.

Nguyen, Q. T., Glasius, M., Sørensen, L. L., Jensen, B., Skov, H., Birmili, W., Wiedensohler, A., Kristensson, A., Nøjgaard, J. K. and Massling, A.: Seasonal variation of atmospheric particle number concentrations, new particle formation and atmospheric oxidation capacity at the high Arctic site Villum Research Station, Station Nord, Atmos. Chem. Phys., 16(17), 11319–11336, doi:10.5194/acp-16-11319-2016, 2016.

Quinn, P. K., Shaw, G., Andrews, E., Dutton, E. G., Ruoho-Airola, T. and Gong, S. L.: Arctic haze: Current trends and knowledge gaps, Tellus, Ser. B Chem. Phys. Meteorol., 59(1), 99–114, doi:10.1111/j.1600-0889.2006.00238.x, 2007.

Sharma, S., Chan, E., Ishizawa, M., Toom-Sauntry, D., Gong, S. L., Li, S. M., Tarasick, D. W., Leaitch, W. R., Norman, A., Quinn, P. K., Bates, T. S., Levasseur, M., Barrie, L. A. and Maenhaut, W.: Influence of transport and ocean ice extent on biogenic aerosol sulfur in the Arctic atmosphere, J. Geophys. Res. Atmos., 117(D12), D12209, doi:10.1029/2011JD017074, 2012.

Tunved, P., Ström, J. and Krejci, R.: Arctic aerosol life cycle: linking aerosol size distributions observed between 2000 and 2010 with air mass transport and precipitation at Zeppelin station, Ny Alesund, Svalbard, Atmos. Chem. Phys, 13, 3643–3660, doi:10.5194/acp-13-3643-2013, 2013.

Wentworth, G. R., Murphy, J. G., Croft, B., Martin, R. V, Pierce, J. R., Côté, J.-S., Courchesne, I., Tremblay, J.-É., Gagnon, J., Thomas, J. L., Sharma, S., Toom-Sauntry, D., Chivulescu, A., Levasseur, M. and Abbatt, J. P. D.: Ammonia in the summertime Arctic marine boundary layer: sources, sinks, and implications, Atmos. Chem. Phys, 16, 1937–1953, doi:10.5194/acp-16-1937-2016, 2016.

Willis, M. D., Burkart, J., Thomas, J. L., Köllner, F., Schneider, J., Bozem, H., Hoor, P. M., Aliabadi, A. A., Schulz, H., Herber, A. B., Leaitch, W. R. and Abbatt, J. P. D.: Growth of nucleation mode particles in the summertime Arctic: a case study, Atmos. Chem. Phys., 16(12), 7663–7679, doi:10.5194/acp-16-7663-2016, 2016.

---

## Referee Comment (RC2) · Anonymous Referee #1 · 5 Oct 2017

This paper presents aerosol size distribution and CCN data from a three week ground-based campaign in the Western Canadian Arctic. The campaign was conducted in the springtime during which it is known that there is a transition from springtime to summertime aerosol conditions. The findings of the paper are that both sets of aerosol distributions (i.e. summer and spring) can be inferred from the data, and the hygroscopicity parameter was observed to be about 0.2. The strength of the paper is that few aerosol measurements have been conducted in the Arctic and so this adds to the data base of such measurements. A weakness is that there is no new conceptual idea presented in the paper.

[Figure]

This paper would represent a more significant contribution if more vertically resolved measurements were presented, and so I am puzzled why the POLAR6 aircraft data are not presented more comprehensively given that the plane was probably flying more frequently that only during the overpass periods. I suggest that these data be added to the revised paper.

Also, the paper should be careful to not claim to have characterized the transition from spring to summer aerosol. That can only be done at a fixed location if a full annual cycle of aerosol parameters is observed, ideally over many years. During a short campaign, the best one can hope to observe are snapshots of different aerosol distributions from different sources. While the authors distinguish their air masses into spring and summer-types, I believe they could just have been easily characterized the periods as continental and marine. For publication, it has to be justified that the general trajectory pattern displayed in Period 2 is indeed characteristic of springtime conditions at this location, i.e. do the size distributions during this period have the character they do because they are more like those in the spring or because they are of continental origin?

Lastly, although the paper is improved compared to the originally submitted version in how it references past work, it is still lacking references to past characterization of the CCN behavior of marine aerosol (currently, only one reference from the authors is presented) and Arctic aerosol. As well, there is a large suite of aerosol size distribution measurements from North American sites (e.g. work by Leaitch et al in Elementa, and Croft et al., Collins et al. and Burkart et al. in ACP) that is ignored and is arguably more relevant than the more geographically distant (but referenced) measurements at Svalbard. Lastly, there is the recent paper by Freud et al. in ACP that comprehensively describes aerosol character across the Arctic. Given that the merit of the current paper is that it adds new measurements to those already performed, it is necessary that the new measurements be presented alongside what has already been reported in the literature. The literature is more comprehensive that what is listed above; I only

included recent publications.

---

## Author Comment (AC1) · 20 Feb 2018

**Author answer to Anonymous Referee #2**

We thank Referee 2 for taking the time to review our manuscript and for giving valuable hints and suggestions. Below, comments from the referee are given in blue while our answers are given in black, with passages including new text given in italic. In the revised version of the text, new text is printed in bold, while text to be deleted is crossed out.

General Comments:

The manuscript by Herenz et al discusses an interpretation of data collected at a ground site in Tuktoyaktuk, Canada, in association with aircraft data during May 2014. This time period is a 'shoulder season' in the Arctic, where the spring 'Arctic Haze' influence is waning and the atmosphere is beginning to take on summer-like characteristics. The paper discusses two types of air mass influences observed at Tuktoyaktuk, which the authors have called 'spring-type' and 'summer-type'. The spring-type air has characteristics similar to Arctic Haze and the summer-type air has an added influence from Aitken mode particles. The measurements and their analysis are technically sound, and data filtering has been presented appropriately. Some of the conclusions and interpretations of the data are confusing or incomplete in the opinion of this reviewer, and have been pointed out below. In addition, the labeling of 'spring-type' and 'summer-type' is perhaps generous (particularly 'summer-type') as it is not shown that the source region or processes shaping the size distribution during the study period are similar to those in the heart of the summer season (July-August). It is perhaps prudent to label the Aitken-mode containing PNSDs as simply 'N. Pacific', since the study does well to describe the highly time-resolved oscillations in air masses during near the spring-summer transition, and correlates this with one of two seasonal maxima in methanesulfonic acid (MSA) (Leaitch et al., 2013; Quinn et al., 2007). As a curious point, the authors performed this research in the Canadian Arctic, yet have not cited much of the existing research that has been conducted in the region, so the suggested or argument-supporting citations provided in this review have been skewed slightly to correct for this fact.

We agree to your general comments, and will elaborate on that more below at your respective remarks. In short, what we did was to change the labeling of the air masses to "accumulation-type" and "Aitken-type" throughout the text. Also, several additional publications are now being cited.

This paper could be accepted to Atmospheric Chemistry and Physics pending major revisions to the discussion and interpretation of the data.

Specific comments:

The abstract of the paper highlights a 'rapid transition from Arctic spring to summer that took place during the measurement period.' This claim is difficult to support directly with the extent of analysis that took presented in this study. Such a claim would be more amenable to inter-seasonal data that are analyzed at high temporal resolution – such a study does not exist to the knowledge of this reviewer – but the fact that two types of air masses influence this site is certainly supported within the manuscript. One longer period of focused analysis in the present manuscript indicates an oscillation between the so-called 'spring' and 'summer' influences on the scale of hours-to-days (Period 1, Figure 6b). Characterizing this oscillating influence as a rapid transition is perhaps technically accurate in some sense; it is also misleading, as the abstract leads the reader to believe that the site will change singularly from 'spring' to 'summer' like a step function. The presented data suggests this is not occurring, and rather the measurement site is alternately influenced by two different source regions (one of which is debatably 'summer-like'), so it is recommended that the abstract be rephrased.

We agree with your comment here and deleted the remark concerning the "rapid transition from Arctic spring to summer". This part of the abstract has been changed to:

*"Basic meteorological parameters and particle number size distributions (PNSDs) were observed and two distinct types of air masses were found."*

The introduction of the paper is quite well written and clear, with some very minor exceptions noted in the detail comments below.

Overall, the measurements presented in the study were analyzed systematically, clearly, and with substantial attention to quantitative detail, including a comprehensive error analysis. Both the size distribution data and the CCN data were used to appropriate lengths, considering uncertainties. The authors should be commended for working with this challenging data set collected in a difficult measurement environment.

We thank the reviewer for these positive and encouraging remarks.

The authors invoke cloud processing as the source of the bi-modal character of the PNSD_C3 size distribution (Page 13, lines 20-24) and this is, indeed, one possibility; however, different explanations that are equally supported by the measurements (as presented) are possible. The interpretation of the factors driving the shape of PNSD_C3 has broader consequences within this study, so this is quite an important point. In addition to the cloud processing argument presented by the authors for the bi-modal character of PNSD_C3, possible explanations include:

1. The PNSD_C3 average size distribution may be a mixture of two types of particles: newly formed particles in the Aitken mode and aged particles in the accumulation mode. Evidence for a mechanism to produce such a scenario has been provided by detailed analysis of size distributions from remote locations in the Arctic (e.g., Collins et al., 2017) wherein new particles may be forming on different days where growth of the existing population occurs simultaneously, causing particles to grow into the accumulation mode size range.

2. The PNSD_C1 mode sizes are quite similar to those of PNSD_C3, and it has been posited that Period 1 has a mixed character of spring-like and summer-like influences. Could there be additional source variability within Period 3 that gives rise to the bimodal character? Indeed on page 17-18, the authors discuss the vertical structure of air masses, suggesting that during Overflight 3, spring-like Arctic air resides above a surface inversion and could mix down into the surface layer. This mechanism would produce a size distribution with characteristics similar to those observed in PNSD_C3.

3. Recent studies at Utqiagvik, AK have shown the strong influence that fossil fuel extraction and processing facilities can have on the aerosol in this region (Gunsch et al., 2017; Kolesar et al., 2017) and could contribute to the shape of PNSD_C3 since the Period 3 trajectories pass over Northern Alaska.

Based on your suggestions we discussed more carefully. However, the bottom line for the observed minimum to occur is that there are at least two types of aerosol with different sources, where accumulation mode particles might usually be assumed to be the more aged ones. For all of the cases you listed, it could still be that particles with larger sizes got a large fraction of their particulate matter by cloud processing, followed by mixing the air mass with a second air mass that contained newly formed particles (as suggested in your point 2). Collins et al. (2017) unfortunately only show one case where there was a particle growth event in parallel to new particle formation and only mention that there were more similar events, while it is also mentioned that the conditions during the measurements often were foggy, so that cloud processing could have occurred in the fog. But still, summarizing your and our arguments, we added to following:

*"While cloud processing is a well known process for gaining particulate matter and growing particles to larger sizes, particles can also grow by generation of particulate matter directly from the gas phase as described recently for Arctic conditions in e.g., Willis et al. (2016), Burkart et al. (2017b) and Collins et al. (2017). The observed minimum in the PNSD occurs when new particle formation takes place, either by adding small particles to an already aged air mass or by mixing of different air masses with one air mass containing aged and the other one newly formed particles, where one could come from aloft. It should also be mentioned that it was recently described in Gunsch et al. (2017) and Kolesar et al. (2017), that emissions from Prudhoe Bay oil field, which is located at the northern shore of Alaska roughly 700 km west of our measurement location, influenced Arctic PNDSs by adding both high concentrations of small particles and*

*particulate mass to larger particles. Summarizing there is a number of reasons that can add to the observed bi-modality of the size distribution, but small, comparably newly formed particles will make up the observed Aitken mode in all cases."*

If uncertainties in the CCN data were smaller, the kappa values of each mode may have provided useful in constraining these four options (one provided by the authors plus three above). Can the authors constrain the system further using any supplementary data or analysis that is not already provided?

We tried all that was possible with the data we have available to get better constrained values, and what is presented here is what possibly can be done. We don't have any supplementary data. Indeed, we regard it as an important result to show the uncertainties in kappa values based on measurement uncertainties. The uncertainty in the CCN measurements and inferred kappa values are based on uncertainties in counting and sizing (size distribution) as well as supersaturation (CCNc). As we used state of the art measurement technology, it can be expected that similar measurement uncertainties will always influence kappa retrievals.

The authors conclude in Section 3.5 that the shape of the PNSD is driven by transport and that local sources are not active in shaping the distribution. This is somewhat antithetic to the behaviour of canonical summertime Arctic aerosol which are thought to have a substantial contribution from new particle formation (Asmi et al., 2016; Croft et al., 2016; Tunved et al., 2013) based on sources within the polar dome via isentropic transport, which the authors discuss. Wet deposition processes (e.g., nucleation and precipitation scavenging) play a major role in shaping the size distribution in summer as well (Browse et al., 2012; Croft et al., 2016), which is not discussed or considered in the manuscript. Recent studies from the Canadian Arctic suggest that transport in heart of the summer is either from the higher Arctic latitudes or from the Canadian boreal forest (Collins et al., 2017; Mungall et al., 2016; Sharma et al., 2012; Wentworth et al., 2016), rather than from the North Pacific.

Critical question: Can the authors provide a stronger connection between the characteristics of the summer Arctic atmosphere and the Period 3-type air masses outside of the presence of the Aitken mode? Should the 'summer-type' (Period 3) air masses be thought of as truly characteristic of summer, or are these specific to a transient seasonal transport pathway from the N. Pacific?

We agree with your argument that Arctic aerosol in summer might differ from that we observed, although the size distributions are similar. Therefore, throughout the whole text, we renamed the types of the air masses, now focusing on what we observe, i.e., they are called accumulation-type and Aitken-type now.

The following was also added to the introduction: "*Indeed, these precipitation related scavenging processes, which are effective from late spring throughout the summer, were shown to be one of the drivers of the yearly cycle in Arctic PNSDs (Browse et al., 2012; Croft et al., 2016a). Resulting low number concentrations of particles in the accumulation mode size range enable new particle formation (NPF).*"

It is also stressed stronger in the text in Section 3.3, that: "*Sources for the precursor gases forming these particles will differ from spring to summer, as mentioned above (Li et al., 1993).*" – where Li et al. (1993) has been cited in the previous version of our manuscript (and still is) as: "The earlier maximum [of MSA] that occurs around the time of our measurements, can be associated with long-range transport from marine source regions from the North Pacific (Li et al., 1993)."

Page 3, lines 14-15: The authors claim a lack of in-situ measurements of CCN in the Arctic region. Recently, CCN and cloud property studies from the NETCARE campaigns have documented aspects of aerosol-cloud interactions, including airborne estimations of activation diameters and chemical effects on CCN activity, specifically in the Canadian Arctic (Burkart et al., 2017; Leaitch et al., 2016; Willis et al., 2016). Other studies, including those referenced in the manuscript, have studied CCN concentrations and properties in other regions (e.g., Moore et al., 2011).

Thank you for pointing out these recent publications to us. We removed the sentence claiming a lack of in-situ measurements and added the following to our text in the paragraph on CCN:

"*Within the NETCARE project based on summer time measurements in the Canadian Arctic Archipelago, high concentrations of newly formed particles were observed particularly in the marine boundary layer and*

*above clouds (Burkart et al., 2017a). One particle growth event measured during NETCARE was described in Willis et al. (2016), showing newly formed particles growing to sizes above 50 nm, subsequently being able to activate to cloud droplets at 0.6% supersaturation. For the same project, Leaitch et al. (2016) examined cloud droplet number concentrations for 62 cloud samples and reported that particles with comparably small diameters, below 50 nm, activated to cloud droplets in 40% of all cases.*"

Page 8, lines 11-13: Since this study was conducted in the Canadian Arctic, it is worthwhile to compare with seasonal aerosol concentrations at Alert, Canada and Utqiagvik (Barrow), AK. If the authors desired to make a more pan-Arctic comparison, inclusion of observations made at Station Nord, Greenland (Nguyen et al., 2016) and Tiksi, Russia (Asmi et al., 2016) should also be discussed.

It is correct that we measured in the Canadian Arctic while the work by Tunved et al. (2013) to which we refer in details was done on Svalbard. We therefore now also added a comparison with literature done in other parts of the Arctic, where also number concentrations and size distribution similar to those observed by us are described. The following two larger paragraphs were added to the introduction, together with a few minor additions:

*"Indeed, these precipitation related scavenging processes, which are effective from late spring throughout the summer, were shown to be one of the drivers of the yearly cycle in Arctic PNSDs (Browse et al., 2012; Croft et al., 2016a). Resulting low number concentrations of particles in the accumulation mode size range enable new particle formation (NPF). The latter is also based on the presence of MSA (methane sulfonic acid), an oxidation product of DMS (dimethyl sulfide) that is emitted by the oceans (Quinn et al., 2007; Leaitch et al., 2013), with increasing emissions related to the decline of the Arctic sea ice cover (Sharma et al., 2012). Additionally, ammonia, also a contributor to NPF, was described to be connected to seabird colonies by Croft et al. (2016b) and Wentworth et al. (2016) and was discussed to have a far ranging influence on the Arctic aerosol."*

*"Croft et al. (2016a) reported data collected in the years 2011 to 2013 from Mt. Zeppelin, i.e., examining different years than Tunved et al. (2013), together with additional data from Alert, Canada. Both yearly cycles of NCN and PNSDs were similar at Alert and Mt. Zeppelin, and also similar to those discussed in Tunved et al. (2013). Croft et al. (2016a) suggest that the observed similarities at these two stations, which are 1000 km apart, and between the different years examined at Mt. Zeppelin indicate the existence of an annual cycle that spans the high Arctic. This assumption is strengthened by Nguyen et al. (2016), reporting again comparable yearly cycles of number concentrations and PNSDs for Villum Research Station in northern Greenland, only differing in more pronounced Aitken modes in the summer month. The shape of the yearly cycle of NCN and the most often occurring PNSDs observed at Tiksi, Russia, described in Asmi et al. (2016), were again similar to those observed at Mt. Zeppelin and Alert. However, number concentrations were higher in general in Tiksi, and NPF events occurred more readily, which is discussed to be related to regional continental sources of nucleating and condensing vapors. Generally, a comparison of PNSDs presented in Freud et al. (2017) from Alert, Villum, Mt. Zeppelin, Tiksi and Barrow (Alaska) shows some differences between Arctic sites due to local effects, but concludes that on a large scale there is a pronounced annual cycle in PNSDs with common features, with all Arctic sites sharing the Asian side as the main large-scale source region of accumulation mode aerosols."*

*"Within the NETCARE project based on summer time measurements in the Canadian Arctic Archipelago, high concentrations of newly formed particles were observed particularly in the marine boundary layer and above clouds (Burkart et al., 2017a). One particle growth event measured during NETCARE was described in Willis et al. (2016), showing newly formed particles growing to sizes above 50 nm, subsequently being able to activate to cloud droplets at 0.6% supersaturation. For the same project, Leaitch et al. (2016) examined cloud droplet number concentrations for 62 cloud samples and reported that particles with comparably small diameters, below 50 nm, activated to cloud droplets in 40% of all cases."*

Page 10, lines 9-10: It is difficult to see the size distribution results as a corroboration of the prior discussion of N_CNraw, since this is the same data shown in a different manner. On page 9, the N_CCN and N_CNraw>150nm were shown and noted as being correlated since the smaller particles are due to pollution.

Figure 5 is simply a re-statement of the same information since the accumulation mode size is 150 nm. It is likely that the choice of 150 nm as a cutoff for the metric in Figure 4 was deliberate, and the data do certainly tell a convincing story about the difference between polluted and 'filtered' data, but these figures represent one piece of evidence, rather than two separate, corroborating pieces of evidence. If other types of techniques distinguished polluted from non-polluted data (chemical or physicochemical observations), then these would corroborate the number size distribution differences between polluted and filtered cases as presented.

We apologize as there is a misunderstanding in what we had intended to say. The corroboration we refer to here was connected to the literature we cited before, i.e., referring to Wegener et al. (2012), who also had seen modes between 10nm and 40nm in median urban PNSD caused by traffic, domestic and district heating, comparable to our results. We rephrased this part to avoid this misunderstanding. It now is:

"*The observations by Wegener et al. (2012) support our assumption …* "

Page 11, lines 19-20: When discussing the Period 3 air masses, no discussion on similarity to prior studies is provided to support the notion that these air masses are indicative of what might be found in the actual summer months at Tuktoyaktuk. On page 13, lines 17-20, a statement about corroborating evidence for N. Pacific sources of air is made in reference to trends in MSA – but it is not shown that this type of air mass is truly typical of Arctic 'summer' as labeled.

We agree, and as mentioned above, we renamed the air masses to accumulation-type and Aitken-type and stress stronger that precursor gasses have different sources and source regions in spring and in summer.

Figure 6: It would be helpful to understand the behavior of the ensemble of back trajectories in the vertical dimension to some degree, so that impacts of surface sources along the trajectory could be understood. Is it possible to include some information on the vertical position of the trajectory or each ensemble of trajectories as a function of time or position?

We agree, that information about the vertical distribution of the air masses could provide further information. To get an idea about the vertical behavior of the back trajectories of the three periods we plotted the pressure along the trajectory in the figure below. For a better overview we only used the trajectories that started right at the location of Tuktoyaktuk (we also started 12 additional trajectories in a close proximity of Tuktoyaktuk per time step to account for uncertainties). It is obvious that the trajectories span a wide vertical range, especially for periods extending further back in time. For Period 1 and 2 one can not really obtain any conclusive information used for further interpretation. Only for period 3 it can be argued that the air masses are in the free troposphere at 100 h back in time, while some of the trajectories got back to the boundary layer (roughly between 150 and 240 h back in time). From this it could be assumed, that the air masses of period three have the closest touch to the ground in the area of the North-West Pacific and Siberia.

We decided not to include this plot in the manuscript as we consider the level of additional information low and inconclusive.

[Figure]

Technical Corrections:

Page 1, line 4: ": : :were indicative for the rapid: : :" change to ": : :were indicative of the rapid: : :"

Page 8, line 11: change to "time periods where the measurements"

Page 9, lines 8-10: These two sentences are hard to follow. It is recommended that the sentences be re-phrased, especially when referring to "it is associated with lower sampling statistics", but the prior sentence refers to two different metrics.

Thank you! The suggested corrections have been made to the manuscript.

References:

Asmi, E., Kondratyev, V., Brus, D., Laurila, T., Lihavainen, H., Backman, J., Vakkari, V., Aurela, M., Hatakka, J., Viisanen, Y., Uttal, T., Ivakhov, V. and Makshtas, A.: Aerosol size distribution seasonal characteristics measured in Tiksi, Russian Arctic, Atmos. Chem. Phys., 16(3), 1271–1287, doi:10.5194/acp-16-1271-2016, 2016.

Browse, J., Carslaw, K. S., Arnold, S. R., Pringle, K. and Boucher, O.: The scavenging processes controlling the seasonal cycle in Arctic sulphate and black carbon aerosol, Atmos. Chem. Phys., 12, 6775–6798, doi:10.5194/acp-12-6775-2012, 2012.

Burkart, J., A. L. Hodshire, E. L. Mungall, J. R. Pierce, D. B. Collins, L. A. Ladino, A. K. Y. Lee, V. Irish, J. J. B. Wentzell, J. Liggio, T. Papakyriakou, J. Murphy, and J. Abbatt: Organic condensation and particle growth to CCN sizes in the summertime marine Arctic is driven by materials more semivolatile than at continental sites, Geophys. Res. Lett., 44, doi:10.1002/2017GL075671, 2017a..

Burkart, J., Willis, M. D., Bozem, H., Thomas, J. L., Law, K., Hoor, P., Aliabadi, A. A., Köllner, F., Schneider, J., Herber, A., Abbatt, J. P. D. and Leaitch, W. R.: Summertime observations of ultrafine particles and cloud condensation nuclei from the boundary layer to the free troposphere in the Arctic, Atmos. Chem. Phys., 17, 5515–5535, doi:10.5194/acp-17-5515-2017, 2017b.

Collins, D. B., Burkart, J., Chang, R. Y.-W., Lizotte, M., Boivin-Rioux, A., Blais, M., Mungall, E. L., Boyer, M., Irish, V. E., Masse, G., Kunkel, D., Tremblay, J.-É., Papakyriakou, T., Bertram, A. K., Bozem, H., Gosselin, M., Levasseur, M. and Abbatt, J. P. D.: Frequent Ultrafine Particle Formation and Growth in the Canadian Arctic Marine Environment, Atmos. Chem. Phys., 2017.

Croft, B., Martin, R. V., Leaitch, W. R., Tunved, P., Breider, T. J., D'Andrea, S. D. and Pierce, J. R.: Processes controlling the annual cycle of Arctic aerosol number and size distributions, Atmos. Chem. Phys., 16(6), 3665–3682, doi:10.5194/acp-16-3665-2016, 2016.

Freud, E., Krejci, R., Tunved, P., Leaitch, R., Nguyen, Q. T., Massling, A., Skov, H., and Barrie, L.: Pan-Arctic aerosol number size distributions: seasonality and transport patterns, Atmos. Chem. Phys., 17, 8101–8128, 2017.

Gunsch, M. J., Kirpes, R. M., Kolesar, K. R., Barrett, T. E., China, S., Sheesley, R. J., Laskin, A., Wiedensohler, A., Tuch, T. and Pratt, K. A.: Contributions of transported Prudhoe Bay oil field emissions to the aerosol population in UtqiaÄa¸vik, Alaska, Atmos. Chem. Phys., 17, 10879–10892, doi:10.5194/acp-17-10879-2017, 2017.

Kolesar, K. R., Cellini, J., Peterson, P. K., Jefferson, A., Tuch, T., Birmili, W., Wiedensohler, A. and Pratt, K. A.: Effect of Prudhoe Bay emissions on atmospheric aerosol growth events observed in UtqiaÄa¸vik (Barrow), Alaska, Atmos. Environ., 152, 146–155, doi:10.1016/j.atmosenv.2016.12.019, 2017.

Leaitch, W. R., Sharma, S., Huang, L., Toom-Sauntry, D., Chivulescu, A., Macdonald, A. M., von Salzen, K., Pierce, J. R., Bertram, A. K., Schroder, J. C., Shantz, N. C., Chang, R. Y.-W. and Norman, A.-L.: Dimethyl sulfide control of the clean summertime Arctic aerosol and cloud, Elem. Sci. Anthr., 1(1), 17, doi:10.12952/journal.elementa.000017, 2013.

Leaitch, W. R., Korolev, A., Aliabadi, A. A., Burkart, J., Willis, M. D., Abbatt, J. P. D., Bozem, H., Hoor, P., Köllner, F., Schneider, J., Herber, A., Konrad, C. and Brauner, R.: Effects of 20–100 nm particles on liquid clouds in the clean summertime Arctic, Atmos. Chem. Phys, 16, 11107–11124, doi:10.5194/acp-16-11107-2016, 2016.

Moore, R. H., Bahreini, R., Brock, C. A., Froyd, K. D., Cozic, J., Holloway, J. S., Middlebrook, A. M., Murphy, D. M. and Nenes, A.: Hygroscopicity and composition of Alaskan Arctic CCN during April 2008, Atmos. Chem. Phys., 11(22), 11807–11825, doi:10.5194/acp-11-11807-2011, 2011.

Mungall, E. L., Croft, B., Lizotte, M., Thomas, J. L., Murphy, J. G., Levasseur, M., Martin, R. V., Wentzell, J. J. B., Liggio, J. and Abbatt, J. P. D.: Dimethyl sulfide in the summertime Arctic atmosphere: measurements and source sensitivity simulations, Atmos. Chem. Phys., 16(11), 6665–6680, doi:10.5194/acp-16-6665-2016, 2016.

Nguyen, Q. T., Glasius, M., Sørensen, L. L., Jensen, B., Skov, H., Birmili, W., Wiedensohler, A., Kristensson, A., Nøjgaard, J. K. and Massling, A.: Seasonal variation of atmospheric particle number concentrations, new particle formation and atmospheric oxidation capacity at the high Arctic site Villum Research Station, Station Nord, Atmos. Chem. Phys., 16(17), 11319–11336, doi:10.5194/acp-16-11319-2016, 2016.

Quinn, P. K., Shaw, G., Andrews, E., Dutton, E. G., Ruoho-Airola, T. and Gong, S. L.: Arctic haze: Current trends and knowledge gaps, Tellus, Ser. B Chem. Phys. Meteorol., 59(1), 99–114, doi:10.1111/j.1600-0889.2006.00238.x, 2007.

Sharma, S., Chan, E., Ishizawa, M., Toom-Sauntry, D., Gong, S. L., Li, S. M., Tarasick, D. W., Leaitch, W. R., Norman, A., Quinn, P. K., Bates, T. S., Levasseur, M., Barrie, L. A. and Maenhaut, W.: Influence of transport and ocean ice extent on biogenic aerosol sulfur in the Arctic atmosphere, J. Geophys. Res. Atmos., 117(D12), D12209, doi:10.1029/2011JD017074, 2012.

Tunved, P., Ström, J. and Krejci, R.: Arctic aerosol life cycle: linking aerosol size distributions observed between 2000 and 2010 with air mass transport and precipitation at Zeppelin station, Ny Alesund, Svalbard, Atmos. Chem. Phys, 13, 3643–3660, doi:10.5194/acp-13-3643-2013, 2013.

Wentworth, G. R., Murphy, J. G., Croft, B., Martin, R. V, Pierce, J. R., Côté, J.-S., Courchesne, I., Tremblay, J.-É., Gagnon, J., Thomas, J. L., Sharma, S., Toom-Sauntry, D., Chivulescu, A., Levasseur, M. and Abbatt, J. P. D.: Ammonia in the summertime Arctic marine boundary layer: sources, sinks, and implications, Atmos. Chem. Phys, 16, 1937–1953, doi:10.5194/acp-16-1937-2016, 2016.

Willis, M. D., Burkart, J., Thomas, J. L., Köllner, F., Schneider, J., Bozem, H., Hoor, P. M., Aliabadi, A. A., Schulz, H., Herber, A. B., Leaitch, W. R. and Abbatt, J. P. D.: Growth of nucleation mode particles in the summertime Arctic: a case study, Atmos. Chem. Phys., 16(12), 7663–7679, doi:10.5194/acp-16-7663-2016, 2016.

---

## Author Comment (AC2) · 20 Feb 2018

**Author answer to Anonymous Referee #1**

We thank Referee 1 for taking the time to review our manuscript and for giving hints and suggestions. Below, comments from the referee are given in blue while our answers are given in black, with passages including new text given in italic. In the revised version of the text, new text is printed in bold, while text to be deleted is crossed out.

This paper presents aerosol size distribution and CCN data from a three week groundbased campaign in the Western Canadian Arctic. The campaign was conducted in the springtime during which it is known that there is a transition from springtime to summertime aerosol conditions. The findings of the paper are that both sets of aerosol distributions (i.e. summer and spring) can be inferred from the data, and the hygroscopicity parameter was observed to be about 0.2. The strength of the paper is that few aerosol measurements have been conducted in the Arctic and so this adds to the data base of such measurements. A weakness is that there is no new conceptual idea presented in the paper.

While currently there is a large amount of publications on Arctic aerosol, most of this (even publications from the last two years of which a number is now additionally included in the manuscript, see literature list below) deals with summer time aerosol and often focuses on new particle formation. Hence, while some data exists also on Arctic aerosol throughout the year, the herein presented data observed in spring, including also directly measured concentrations of CCN and the particle hygroscopicity parameter derived from measurements, adds new data to the still scarce database. Similarly, the comparison of simultaneously taken ground based and airborne measurements, presented for all overflights made during the campaign, adds new and valuable information to the discussion concerning the connection between ground based and airborne conditions. Also, the uncertainty analysis concerning the hygroscopicity parameters, which shows limitations of what can be learned by the respective kind of data, is new and valuable on its own, beyond its applicability to Arctic aerosol.

This paper would represent a more significant contribution if more vertically resolved measurements were presented, and so I am puzzled why the POLAR6 aircraft data are not presented more comprehensively given that the plane was probably flying more frequently that only during the overpass periods. I suggest that these data be added to the revised paper.

The work presented here is based on the measurements done by the main author who, together with some of the co-authors, is a member of the TROPOS cloud group. The TROPOS cloud groups' role during RACEPAC was to contribute ground based measurements, and these are analyzed herein, together with some aircraft data that others were willing to share with us and that were taken by these other groups during times of overflights.

The idea of the ground based measurements was to have a continuous characterization of the aerosol on ground, as such a continuous record naturally cannot be obtained by aircraft measurements. Additionally, in this study, all existing size distribution data taken during overflights were used to compare ground based data with aircraft data. Other aircraft data will be published by others and a further inclusion of them is beyond the scope of this work.

Also, the paper should be careful to not claim to have characterized the transition from spring to summer aerosol. That can only be done at a fixed location if a full annual cycle of aerosol parameters is observed, ideally over many years. During a short campaign, the best one can hope to observe are snapshots of different aerosol distributions from different sources. While the authors distinguish their air masses into spring and summer-types, I believe they could just have been easily characterized the periods as continental and marine. For publication, it has to be justified that the general trajectory pattern displayed in Period 2 is

indeed characteristic of springtime conditions at this location, i.e. do the size distributions during this period have the character they do because they are more like those in the spring or because they are of continental origin?

Thank you for pointing this out. We agree and have changed the naming of the two types of aerosol to "accumulation-type" and "Aitken-type" throughout the manuscript and also changed the wording in the text concerning the "transition" accordingly where needed.

The size distribution of Period 2, which is mono-modal and of "accumulation-type" is typical for the Arctic spring aerosol (shape and integrated number) and compares well with size distributions measured by others at different Arctic sites during this season e.g., Freud et al. (2017), Tunved et al. (2013). The trajectories of Period 2 show the advection of air masses with the origin in the central arctic rather than the North American continental region. These facts, i.e., shape and integrated number of the size distribution as well as the origin of the air masses, are clear indicators for the presence of air masses that are typical for the Arctic during spring. The main characteristic of the Arctic aerosol during spring is that most significant sources are inside the Arctic dome (e.g., biomass burning and industry at the Asian continent) and particles have a long residence time due to a lack of precipitation, and released aerosol particles are distributed widespread over the Arctic. To make this clearer in the manuscript, we have added this passage in the introduction:

*"Croft et al. (2016a) reported data collected in the years 2011 to 2013 from Mt. Zeppelin, i.e., examining different years than Tunved et al. (2013), together with additional data from Alert, Canada. Both yearly cycles of NCN and PNSDs were similar at Alert and Mt. Zeppelin, and also similar to those discussed in Tunved et al. (2013). Croft et al. (2016a) suggest that the observed similarities at these two stations, which are 1000 km apart, and between the different years examined at Mt. Zeppelin indicate the existence of an annual cycle that spans the high Arctic. This assumption is strengthened by Nguyen et al. (2016), reporting again comparable yearly cycles of number concentrations and PNSDs for Villum Research Station in northern Greenland, only differing in more pronounced Aitken modes in the summer month. The shape of the yearly cycle of NCN and the most often occurring PNSDs observed at Tiksi, Russia, described in Asmi et al. (2016), were again similar to those observed at Mt. Zeppelin and Alert. However, number concentrations were higher in general in Tiksi, and NPF events occurred more readily, which is discussed to be related to regional continental sources of nucleating and condensing vapors. Generally, a comparison of PNSDs presented in Freud et al. (2017) from Alert, Villum, Mt. Zeppelin, Tiksi and Barrow (Alaska) shows some differences between Arctic sites due to local effects, but concludes that on a large scale there is a pronounced annual cycle in PNSDs with common features, with all Arctic sites sharing the Asian side as the main large-scale source region of accumulation mode aerosols."*

Also we added a new section showing the results of the Potential Source Contribution Function (PSCF) to identify regions contributing to high $N_{CN}$ measurements. Even though this analysis is not done specifically for Period 2, it shows that especially the former mentioned regions in Siberia ("Asian site") may significantly influence the aerosol at Tuktoyaktuk.

Lastly, although the paper is improved compared to the originally submitted version in how it references past work, it is still lacking references to past characterization of the CCN behavior of marine aerosol (currently, only one reference from the authors is presented) and Arctic aerosol. As well, there is a large suite of aerosol size distribution measurements from North American sites (e.g. work by Leaitch et al in Elementa, and Croft et al., Collins et al. and Burkart et al. in ACP) that is ignored and is arguably more relevant than the more geographically distant (but referenced) measurements at Svalbard. Lastly, there is the recent paper by Freud et al. in ACP that comprehensively describes aerosol character across the Arctic. Given that the merit of the current paper is that it adds new measurements to those already performed, it is necessary that the new measurements be presented alongside what has already been reported in the literature. The literature is more comprehensive that what is listed above; I only included recent publications.

We included an additional list of publications and hope this suffices your requirements. There is certainly more, however, as this work is not a review, adding more is beyond the scope of this work. To account for your suggestions we added the following paragraphs to

the introduction:

*"Indeed, these precipitation related scavenging processes, which are effective from late spring throughout the summer, were shown to be one of the drivers of the yearly cycle in Arctic PNSDs (Browse et al., 2012; Croft et al., 2016a). Resulting low number concentrations of particles in the accumulation mode size range enable new particle formation (NPF). The latter is also based on the presence of MSA (methane sulfonic acid), an oxidation product of DMS (dimethyl sulfide) that is emitted by the oceans (Quinn et al., 2007; Leaitch et al., 2013), with increasing emissions related to the decline of the Arctic sea ice cover (Sharma et al., 2012). Additionally, ammonia, also a contributor to NPF, was described to be connected to seabird colonies by Croft et al. (2016b) and Wentworth et al. (2016) and was discussed to have a far ranging influence on the Arctic aerosol."*

*"Croft et al. (2016a) reported data collected in the years 2011 to 2013 from Mt. Zeppelin, i.e., examining different years than Tunved et al. (2013), together with additional data from Alert, Canada. Both yearly cycles of NCN and PNSDs were similar at Alert and Mt. Zeppelin, and also similar to those discussed in Tunved et al. (2013). Croft et al. (2016a) suggest that the observed similarities at these two stations, which are 1000 km apart, and between the different years examined at Mt. Zeppelin indicate the existence of an annual cycle that spans the high Arctic. This assumption is strengthened by Nguyen et al. (2016), reporting again comparable yearly cycles of number concentrations and PNSDs for Villum Research Station in northern Greenland, only differing in more pronounced Aitken modes in the summer month. The shape of the yearly cycle of NCN and the most often occurring PNSDs observed at Tiksi, Russia, described in Asmi et al. (2016), were again similar to those observed at Mt. Zeppelin and Alert. However, number concentrations were higher in general in Tiksi, and NPF events occurred more readily, which is discussed to be related to regional continental sources of nucleating and condensing vapors. Generally, a comparison of PNSDs presented in Freud et al. (2017) from Alert, Villum, Mt. Zeppelin, Tiksi and Barrow (Alaska) shows some differences between Arctic sites due to local effects, but concludes that on a large scale there is a pronounced annual cycle in PNSDs with common features, with all Arctic sites sharing the Asian side as the main large-scale source region of accumulation mode aerosols."*

*"Within the NETCARE project based on summer time measurements in the Canadian Arctic Archipelago, high concentrations of newly formed particles were observed particularly in the marine boundary layer and above clouds (Burkart et al., 2017a). One particle growth event measured during NETCARE was described in Willis et al. (2016), showing newly formed particles growing to sizes above 50 nm, subsequently being able to activate to droplets at 0.6% supersaturation. For the same project, Leaitch et al. (2016) examined cloud droplet number concentrations for 62 cloud samples and reported that particles with comparably small diameters, below 50 nm, activated to cloud droplets in 40% of all cases."*

To the size distribution section:

*"While cloud processing is a well known process for gaining particulate matter and growing particles to larger sizes, particles can also grow by generation of particulate matter directly from the gas phase as described recently for Arctic conditions in e.g., Willis et al. (2016), Burkart et al. (2017b) and Collins et al. (2017). In general, the observed minimum in the PNSD occurs when new particle formation takes place, either by adding small particles to an already aged air mass or by mixing of different air masses with one air mass containing aged and the other one newly formed particles, where one could come from aloft. It should also be mentioned that it was recently described in Gunsch et al. (2017) and Kolesar et al. (2017), that emissions from Prudhoe Bay oil field, which is located at the northern shore of Alaska roughly 700 km west of our measurement location, influenced Arctic PNDSs by adding both high concentrations of small particles and particulate mass to larger particles. Summarizing there is a number of reasons that can add to the observed bi-modality of the size distribution, but small, comparably newly formed particles will make up the observed Aitken mode in all cases."*

and to the CCN section:

*"Within the NETCARE project based on summer time measurements in the Canadian Arctic Archipelago, high concentrations of newly formed particles were observed particularly in the marine boundary layer and above clouds (Burkart et al., 2017a). One particle growth event measured during NETCARE was*

*described in Willis et al. (2016), showing newly formed particles growing to sizes above 50 nm, subsequently being able to activate to droplets at 0.6% supersaturation. For the same project, Leaitch et al. (2016) examined cloud droplet number concentrations for 62 cloud samples and reported that particles with comparably small diameters, below 50 nm, activated to cloud droplets in 40% of all cases.*"

Here you find the list of references, that were added to the manuscript to respond of your suggestions:

AMAP: AMAP Assessment 2006: Acidifying Pollutants, Arctic Haze, and Acidification in the Arctic., Arctic Monitoring and Assessment Programme (AMAP), 2006.

Ashbaugh, L. L., Malm, W. C., and Sadeh, W. Z.: A residence time probability analysis of sulfur concentrations at Grand Canyon National Park, Atmos. Environ., 19, 1263 – 1270, 1985.

Asmi, E., Kondratyev, V., Brus, D., Laurila, T., Lihavainen, H., Backman, J., Vakkari, V., Aurela, M., Hatakka, J., Viisanen, Y., Uttal, T., Ivakhov, V. and Makshtas, A.: Aerosol size distribution seasonal characteristics measured in Tiksi, Russian Arctic, Atmos. Chem. Phys., 16(3), 1271–1287, doi:10.5194/acp-16-1271-2016, 2016.

Browse, J., Carslaw, K. S., Arnold, S. R., Pringle, K. and Boucher, O.: The scavenging processes controlling the seasonal cycle in Arctic sulphate and black carbon aerosol, Atmos. Chem. Phys., 12, 6775–6798, doi:10.5194/acp-12-6775-2012, 2012.

Burkart, J., A. L. Hodshire, E. L. Mungall, J. R. Pierce, D. B. Collins, L. A. Ladino, A. K. Y. Lee, V. Irish, J. J. B. Wentzell, J. Liggio, T. Papakyriakou, J. Murphy, and J. Abbatt: Organic condensation and particle growth to CCN sizes in the summertime marine Arctic is driven by materials more semivolatile than at continental sites, Geophys. Res. Lett., 44, doi:10.1002/2017GL075671, 2017a.

Burkart, J., Willis, M. D., Bozem, H., Thomas, J. L., Law, K., Hoor, P., Aliabadi, A. A., Köllner, F., Schneider, J., Herber, A., Abbatt, J. P. D. and Leaitch, W. R.: Summertime observations of ultrafine particles and cloud condensation nuclei from the boundary layer to the free troposphere in the Arctic, Atmos. Chem. Phys., 17, 5515–5535, doi:10.5194/acp-17-5515-2017, 2017b.

Collins, D. B., Burkart, J., Chang, R. Y.-W., Lizotte, M., Boivin-Rioux, A., Blais, M., Mungall, E. L., Boyer, M., Irish, V. E., Masse, G., Kunkel, D., Tremblay, J.-É., Papakyriakou, T., Bertram, A. K., Bozem, H., Gosselin, M., Levasseur, M. and Abbatt, J. P.

D.: Frequent Ultrafine Particle Formation and Growth in the Canadian Arctic Marine Environment, Atmos. Chem. Phys., 2017.

Croft, B., Martin, R. V., Leaitch, W. R., Tunved, P., Breider, T. J., D'Andrea, S. D. and Pierce, J. R.: Processes controlling the annual cycle of Arctic aerosol number and size distributions, Atmos. Chem. Phys., 16(6), 3665–3682, doi:10.5194/acp-16-3665-2016, 2016.

Gunsch, M. J., Kirpes, R. M., Kolesar, K. R., Barrett, T. E., China, S., Sheesley, R. J., Laskin, A., Wiedensohler, A., Tuch, T. and Pratt, K. A.: Contributions of transported Prudhoe Bay oil field emissions to the aerosol population in UtqiaÄa vik, Alaska, Atmos. Chem. Phys., 17, 10879–10892, doi:10.5194/acp-17-10879-2017, 2017.

Dall'Osto, M., Ovadnevaite, J., Paglione, M., Beddows, D. C., Ceburnis, D., Cree, C., Cortés, P., Zamanillo, M., Nunes, S. O., Pérez, G. L., et al.: Antarctic sea ice region as a source of biogenic organic nitrogen in aerosols, Sci. Rep., 7, 6047, 2017b.

Fleming, Z. L., Monks, P. S., and Manning, A. J.: Review: Untangling the influence of air-mass history in interpreting observed atmospheric composition, Atmos. Res., 104-105, 1 – 39, 2012.

Freud, E., Krejci, R., Tunved, P., Leaitch, R., Nguyen, Q. T., Massling, A., Skov, H., and Barrie, L.: Pan-Arctic aerosol number size distributions: seasonality and transport patterns, Atmos. Chem. Phys., 17, 8101–8128, 2017.

Gunsch, M. J., Kirpes, R. M., Kolesar, K. R., Barrett, T. E., China, S., Sheesley, R. J., Laskin, A., Wiedensohler, A., Tuch, T. and Pratt, K. A.: Contributions of transported Prudhoe Bay oil field emissions to the aerosol population in UtqiaÄa vik, Alaska, Atmos. Chem. Phys., 17, 10879–10892, doi:10.5194/acp-17-10879-2017, 2017.

Kolesar, K. R., Cellini, J., Peterson, P. K., Jefferson, A., Tuch, T., Birmili, W., Wiedensohler, A. and Pratt, K. A.: Effect of Prudhoe Bay emissions on atmospheric aerosol growth events observed in UtqiaÄa vik (Barrow), Alaska, Atmos. Environ., 152, 146–155, doi:10.1016/j.atmosenv.2016.12.019, 2017.

Leaitch, W. R., Sharma, S., Huang, L., Toom-Sauntry, D., Chivulescu, A., Macdonald, A. M., von Salzen, K., Pierce, J. R., Bertram, A. K., Schroder, J. C., Shantz, N. C., Chang, R. Y.-W. and Norman, A.-L.: Dimethyl sulfide control of

the clean summertime Arctic aerosol and cloud, Elem. Sci. Anthr., 1(1), 17, doi:10.12952/journal.elementa.000017, 2013.

Leaitch, W. R., Korolev, A., Aliabadi, A. A., Burkart, J., Willis, M. D., Abbatt, J. P. D., Bozem, H., Hoor, P., Köllner, F., Schneider, J., Herber, A., Konrad, C. and Brauner, R.: Effects of 20–100 nm particles on liquid clouds in the clean summertime Arctic, Atmos. Chem. Phys, 16, 11107–11124, doi:10.5194/acp-16-11107-2016, 2016.

Nguyen, Q. T., Glasius, M., Sørensen, L. L., Jensen, B., Skov, H., Birmili, W., Wiedensohler, A., Kristensson, A., Nøjgaard, J. K. and Massling, A.: Seasonal variation of atmospheric particle number concentrations, new particle formation and atmospheric oxidation capacity at the high Arctic site Villum Research Station, Station Nord, Atmos. Chem. Phys., 16(17), 11319–11336, doi:10.5194/acp-16-11319-2016, 2016.

Sharma, S., Chan, E., Ishizawa, M., Toom-Sauntry, D., Gong, S. L., Li, S. M., Tarasick, D. W., Leaitch, W. R., Norman, A., Quinn, P. K., Bates, T. S., Levasseur, M., Barrie, L. A. and Maenhaut, W.: Influence of transport and ocean ice extent on biogenic aerosol sulfur in the Arctic atmosphere, J. Geophys. Res. Atmos., 117(D12), D12209, doi:10.1029/2011JD017074, 2012.

Waked, A., Favez, O., Alleman, L. Y., Piot, C., Petit, J.-E., Delaunay, T., Verlinden, E., Golly, B., Besombes, J.-L., Jaffrezo, J.-L., and Leoz-Garziandia, E.: Source apportionment of PM10 in a north-western Europe regional urban background site (Lens, France) using positive matrix factorization and including primary biogenic emissions, Atmos. Chem. Phys., 14, 3325–3346, 2014.

Wentworth, G. R., Murphy, J. G., Croft, B., Martin, R. V, Pierce, J. R., Côté, J.-S., Courchesne, I., Tremblay, J.-É., Gagnon, J., Thomas, J. L., Sharma, S., Toom-Sauntry, D., Chivulescu, A., Levasseur, M. and Abbatt, J. P. D.: Ammonia in the summertime Arctic marine boundary layer: sources, sinks, and implications, Atmos. Chem. Phys, 16, 1937–1953, doi:10.5194/acp-16-1937-2016, 2016.

Willis, M. D., Burkart, J., Thomas, J. L., Köllner, F., Schneider, J., Bozem, H., Hoor, P. M., Aliabadi, A. A., Schulz, H., Herber, A. B., Leaitch, W. R. and Abbatt, J. P. D.: Growth of nucleation mode particles in the summertime Arctic: a case study, Atmos. Chem. Phys., 16(12), 7663–7679, doi:10.5194/acp-16-7663-2016, 2016.

Yli-Tuomi, T., Hopke, P. K., Paatero, P., Basunia, M., Landsberger, S., Viisanen, Y., and Paatero, J.: Atmospheric aerosol over Finnish Arctic: source analysis by the multilinear engine and the potential source contribution function, Atmos. Environ., 37, 4381 – 4392, 2003.

---

## Author Comment (AC3) · 20 Feb 2018

[revised manuscript text omitted]

During the transition from spring to summer an increased vertical mixing causes the presence of low-level clouds, and the related wet removal stops the Arctic haze period (Tunved et al., 2013), making the well aged (Heintzenberg, 1980) Arctic haze particles of the accumulation mode disappear (Engvall et al., 2008; Tunved et al., 2013).  **Indeed, these precipitation related scavenging processes, which are effective from late spring throughout the summer, were shown to be among the drivers of the yearly cycle in Arctic PNSDs (Browse et al., 2012; Croft et al., 2016a). Resulting low number concentra-**

tions of particles in the accumulation mode size range enable new particle formation (NPF). The latter is also based on the presence of MSA (methane sulfonic acid), an oxidation product of DMS (dimethyl sulfide) that is emitted due to biological activity in the oceans (Quinn et al., 2007; Leaitch et al., 2013), with increasing emissions related to the decline of the Arctic sea ice cover (Sharma et al., 2012). Additionally, ammonia, also a contributor to NPF, was described to be connected to seabird colonies by Croft et al. (2016b) and Wentworth et al. (2016) and was discussed to have a far ranging influence on the Arctic aerosol. In general, during the Arctic summer locally and freshly produced aerosol particle species are dominant, driven by an increase in both biological activity and photochemistry, (Ström et al., 2009) showing up as a pronounced Aitken mode in PNSDs in summer month, particularly in July and August (Tunved et al., 2013).

Consequently, the Arctic aerosol particle number size distribution as well as the particle number concentrations show a large seasonal variability (Tunved et al., 2013). Moreover the sources and sinks for Arctic aerosol particles are subject to the fast changes in the Arctic that currently take place. Dall'Osto et al. (2017a) for instance found a negative correlation between the Arctic sea ice extent and NPF events, that were observed at Mt Zeppelin (Svalbard). From this connection follows an increased new particle production due to the current decrease in the sea ice pack extent (Dall'Osto et al., 2017a).

Croft et al. (2016a) reported data collected in the years 2011 to 2013 from Mt. Zeppelin, i.e., examining different years than Tunved et al. (2013), together with additional data from Alert, Canada. Both yearly cycles of $N_{CN}$ and PNSDs were similar at Alert and Mt. Zeppelin, and also similar to those discussed in Tunved et al. (2013). Croft et al. (2016a) suggest that the observed similarities at these two stations, which are 1000 km apart, and between the different years examined at Mt. Zeppelin indicate the existence of an annual cycle that spans the high Arctic. This assumption is corroborated by Nguyen et al. (2016), reporting comparable yearly cycles of number concentrations and PNSDs from the Villum Research Station in northern Greenland, only differing in more pronounced Aitken modes in the summer months. The shape of the yearly cycle of $N_{CN}$ and the most often occurring PNSDs observed in Tiksi, Russia, described in Asmi et al. (2016), were again similar to those observed at Mt. Zeppelin and Alert. However, number concentrations were higher in general in Tiksi, and NPF events occurred more readily, which is suggested to be related to regional continental sources of nucleating and condensing vapors. Generally, a comparison of PNSDs presented in Freud et al. (2017) from Alert, Villum Research Station, Mt. Zeppelin, Tiksi and Barrow (Alaska) shows some differences between Arctic sites due to local effects, but indicates that on a large scale there is a pronounced annual cycle in PNSDs with common features, with all Arctic sites sharing the Asian continent as the main large-scale source region of accumulation mode aerosols.

Similarly, also the Arctic CCN number concentrations vary, with values between less than $100 \mathrm{cm}^{-3}$ (pristine Arctic background), occasionally less than $1 \mathrm{cm}^{-3}$ (Mauritsen et al., 2011), and up to $1000 \mathrm{cm}^{-3}$ (in Arctic haze layers, Moore et al. (2011) and references therein). In the previously mentioned study by Dall'Osto et al. (2017a) it is also shown that the NPF events and the growth of these aerosol particles to a larger size can affect the CCN number concentration. Dall'Osto et al. (2017a) found an increase of the CCN number concentration (measured at a super saturation of $0.4\,\%$) of $21\,\%$ which is linked to NPF events. Within the NETCARE project based on summer time measurements in the Canadian Arctic Archipelago, high concentrations of newly formed particles were observed particularly in the marine boundary layer and above clouds (Burkart et al., 2017a). One particle growth event measured during NETCARE was described in Willis et al. (2016), showing

**newly formed particles growing to sizes above 50 nm, subsequently being able to activate to cloud droplets at 0.6%**
**supersaturation. For the same project, Leaitch et al. (2016) examined cloud droplet number concentrations for 62 cloud**
**samples and reported that particles with comparably small diameters, below 50 nm, activated to cloud droplets in 40%**
**of all cases.**

5   Besides the fact that aerosol particles need to have a certain size to act as CCN, also the aerosol particle chemistry matters in terms of the activation to a cloud droplet. The single hygroscopicity parameter $\kappa$ (Petters and Kreidenweis, 2007) is commonly used to express the affinity of aerosol particles to water and characterizes their CCN activity. The hygroscopicity of the Arctic aerosol particulate matter (PM) was also found to show a seasonality. $\kappa$ values determined from CCN measurements done on water soluble particulate matter collected in Spitzbergen by Silvergren et al. (2014) were between 0.3 and 0.7, with a minimum

10  from March to May and a maximum in October. The past and future changes in the Arctic climate may cause changes of CCN number concentrations and their properties and consequently also to the sources and sinks of Arctic CCN. Hence there is a need for measurements in the Arctic region to quantify the CN and CCN number concentrations, their sources and sinks as well as the aerosol particle hygroscopicity.

[revised manuscript text omitted]

**3.2 Identification of air mass origins and potential source regions**

**We applied two approaches to investigate the history of the measured air masses. First, the origin of the air masses of the three periods is identified by means of air mass back trajectories. Second, the Potential Source Contribution Function (PSCF), which is a residence time analysis that results in a probability field (Fleming et al., 2012), is applied to identify regions that potentially act as source regions for aerosol particles measured throughout the whole campaign.**

**3.2.1 Origin of sampled air masses of the three periods**

To assess the origin of the air masses  **of the three periods** we used the Lagrangian analysis tool (Sprenger and Wernli, 2015). The LAGRANTO backward trajectories were calculated based on analysis data

from the European Center of Medium-Range Weather Forecasts (ECMWF). The data used have a horizontal grid spacing of $0.5°$ and 137 vertical hybrid sigma-pressure levels from the surface up to $0.01\,\mathrm{hPa}$. Hourly 10-day back trajectories were started in the region around Tuktoyaktuk (69.43°N, 133.00°W) at a pressure level of $25\,\mathrm{hPa}$ below surface pressure. More specifically, we initialized trajectories at 13 receptor sites in the horizontal plane, accounting for the uncertainty introduced due to the relatively coarse horizontal grid and the release at an individual point. One receptor site was directly at the coordinates of the measurement station and 12 around the station. The upper panel of Figure 6 depicts the bundle of trajectories for the three time periods.

[Figure]

**Figure 6.** a) 10 day back trajectories of the three periods started in and around Tuktoyaktuk at a pressure level of $25\,\mathrm{hPa}$ above the surface pressure. b) The total number of 13 trajectories per hour of Period 1 was split up into the number of trajectories that came from east ($N_{\underline{spring-type}\mathbf{accumulation}}$) or west ($N_{\underline{summer-type}\mathbf{Aitken}}$) to illustrate the alternation of the air mass origin.

Two main air mass origins were observed. The air masses of Period 2 originated in the north east of Canada (in the province Nunavut). During May air masses from this area are typically highly contaminated due to high winter and springtime aerosol particle burdens which can be observed all over the Arctic (Shaw, 1995). In the following these air masses are termed "**accumulation**-type air mass". The air masses of Period 3 originated in the southwest of Tuktoyaktuk (Eastern Russia, Kamchatka and the unfrozen North Pacific). In the following, these air masses are named "**Aitken**-type air mass", **and the naming of the air masses will be explained in the next section, Section 3.3.** Further the trajectories indicate that Period 1 includes both, **accumulation**- and **Aitken**-type air masses. The distribution of these two air masses during Period 1 is visible in the lower panel of Figure 6. The figure shows the number of trajectories per hour (for Period 1) that originated east ($N_{\underline{spring-type}\textbf{accumulation}}$) or west ($N_{\underline{summer-type}\textbf{Aitken}}$) of Tuktoyaktuk. It can be seen that during the

first part of Period 1 (till 16:00 of 04 May 2014) the air masses in Tuktoyaktuk were a mixture of **accumulation**- and **Aitken**-type air masses. Until the 3rd of May Tuktoyaktuk was influenced by an anticyclone with a maximum pressure of 1035 hPa. The low pressure gradient of this anticyclone led to a low wind velocity and a baffling wind (see lower panel of Figure 3) which caused an alternation between the two air mass origins at the beginning of Period 1. The second part of Period 1 is characterized by a decreasing surface pressure and a constant easterly wind. With a temporal shift of less than one day also

$N_{\underline{summer-type}\textbf{Aitken}}$ decreased which indicates that only **accumulation**-type air masses were present at Tuktoyaktuk.

**3.2.2 PSCF analysis**

**The Potential Source Contribution Function (PSCF) is a receptor modeling method, that is based on air mass back trajectories. Originally it was developed by Ashbaugh et al. (1985) and was applied in numerous studies, also high latitude studies e.g., Dall'Osto et al. (2017b) (Antarctic) and Yli-Tuomi et al. (2003) (Arctic), before. This model is commonly used to identify regions that have the potential to contribute to high values of measured concentrations at a receptor site. In the present study, the NOAA HYSPLIT trajectory model was used to calculate hourly resolved 10-day back trajectories based on 1x1° gdas meteorological data. To account for uncertainties in back trajectory analysis, every hour a set of 15 back trajectories was calculated, which is composed of 5 different plane locations (one exactly at the measurement station and 4 in close proximity around it) at three altitudes (100m, 200m and 300m above the surface level). We apply the PSCF model according to Hopke (2016) on hourly average $N_{CN}$ values with the 75% percentile as threshold value. The threshold value defines, which $N_{CN}$ value is considered as a high concentration. We calculated the PSCF on the basis of 5x5° grid cells. To account for bad statistics in grid cells with a low trajectory density, a weighting function according to Waked et al. (2014) was applied.**

**The spatial distribution of the PSCF of $N_{CN}$ is mapped in Figure 7. The map shows several areas of enhanced PSCF values, that can be linked to potential source regions. One of these spots of enhanced PSCF values is located in Central Canada, which potentially can be linked to biomass burning. To proof this possible connection, we used the MODIS Active Fire Product (MCD14ML) to display the active wild fire spots that occurred between April 21th and May 17th in 2014 as magenta dots in the map. Due to the local proximity of these enhanced PSCF values and a spot of detected active fires south of them, it is possible, that we measured high $N_{CN}$ values due to biomass burning in Central Canada.**

[Figure]

**Figure 7.** Map showing the PSCF function of $N_{CN}$, calculated on the basis of concentrations exceeding the 75% percentile. The colorbar indicates the PSCF value and the red dot, the diamond and the square indicate the position of Tuktoyaktuk, Prudhoe Bay (Alaska, USA) and Norilsk (Russia), respectively. The purple dots display the location of active fire spots that occurred between April 21th and May 17th in 2014, detected by MODIS (Active Fire Product - MCD14ML).

Also active fires detected in Siberia show coincidence with a spot of enhanced PSCF values. From a detailed analysis on a daily basis follows, that especially this region of high PSCF values shows the occurrence of active fires throughout the whole time of analysis. Another spot in Siberia can be explained due to its proximity to Norilsk (red square in Figure 7) which is considered to be an Arctic point source for emissions due to nickel mining and aluminum plants. Although, improvement has been achieved since the' 80s, Norilsk still remains one of the largest source of anthropogenic Arctic air pollution, mainly due to the emission of particulates and sulfur dioxide (AMAP, 2006). Another point source of anthropogenic Arctic emissions is the Prudhoe Bay Oil Field in North Alaska, marked with a red diamond in Figure 7. As already mentioned in the introduction, Gunsch et al. (2017) and Kolesar et al. (2017) recently found the emissions of Prudhoe Bay Oil Field to cause increased $N_{CN}$ values (13-746nm) and have impact on growth events of nucleation and Aitken mode aerosol particles. Our PSCF analysis results in a spot of increased PSCF values in the vicinity of Prudhoe

**Bay Oil Field.** This indicates that Prudhoe Bay Oil Field emissions potentially lead to enhanced $N_{CN}$ values measured in Tuktoyaktuk. The largest area of enhanced PSCF values is the area of the North-West Pacific. This region seems to overall cause relatively high $N_{CN}$ values measured at Tuktoyaktuk, most likely due to marine emissions. A detailed discussion about the impact of marine emissions on the aerosol particles measured in Tuktoyaktuk is presented in the following section.

The above discussed PSCF results give a rough idea about the location of possible aerosol particle sources for our measurements at Tuktoyaktuk. However, in our case, the precision of the PSCF method is limited due to the small amount of data.

**3.3 $PNSD$ of the three periods**

Figure 8a shows the median of the $PNSD_c$ of the three periods discussed in Section 3.2.1, together with the 25 % and 75 % percentiles. They were computed to examine their dependence on the origin of the air mass. The $PNSD_c$ of Period 1 ($PNSD_{c1}$) and Period 3 ($PNSD_{c3}$) are bi-modal with an Aitken mode below 100 nm and an accumulation mode above 100 nm whereas that of Period 2 ($PNSD_{c2}$) only shows the accumulation mode. The large variability we observed in the shape of the $PNSD$ is typical for the transition period from Arctic spring to summer, i.e. higher variation of source regions during the Arctic summer.

**As described in more detail in the introduction, although there is a pronounced annual cycle in PNSDs in the Arctic, common features concerning PNSDs are shared across the Arctic (Freud et al., 2017), and we use PNSDs reported in Tunved et al. (2013) for comparison with our data in the following, as these were the first long term data describing the annual cycle.**

$PNSD_{c2}$ is similar to the $PNSD$ that Tunved et al. (2013) observed for March and April on Svalbard. A direct comparison of $PNSD_{c2}$ and the median April $PNSD$ of Tunved et al. (2013) is shown in Figure 8b. The mono-modal accumulation mode aerosol is characteristic for the Arctic haze which mainly consists of particulate organic matter (POM) and sulfate (Quinn et al., 2002). Single particle analysis of aerosol particles samples taken at the Zeppelin Station, Svalbard, that occurred before the transition to the Arctic summer showed a dominance of spherical organic like particles in the submicrometer range with an Eurasian influence (Behrenfeld et al., 2008). These Arctic haze aerosol particles typically are well aged (Heintzenberg, 1980; Quinn et al., 2002). Due to the shape of $PNSD_{c2}$, and since the air mass of Period 2 has its origin in a region where conditions in May are still winterly, it is very likely that we observed a typical Arctic haze air mass. In contrast, $PNSD_{c3}$ is comparable to $PNSD$s that are reported by Tunved et al. (2013) for June and July. $PNSD_{c3}$ and the median June $PNSD$ of Tunved et al. (2013) are depicted in Figure 8b. In addition to the accumulation mode the bi-modal summer time Arctic $PNSD$ shows an Aitken mode which most likely originates from particles formed by new particle formation (Engvall et al., 2008; Wiedensohler et al., 2011). A common precursor gas for new particle formation is dimethylsulfide (DMS) emitted from oceanic phytoplankton. This precursor is known to be more abundant during the Arctic summer when the marine biological activity has its maximum. An indicator for the presence of DMS is its oxidation product methanesulfonic acid (MSA) (Quinn et al., 2007). MSA also could be directly detected as component of the particulate matter itself in remote marine background aerosol and in plankton bloom areas (Zorn et al., 2008). Quinn et al. (2007) report the concentration of MSA for several Arctic

measurement stations (e.g. Barrow and Alert - Tuktoyaktuk is located between the two) during at least 7 years. The MSA concentration starts to increase in April and has two maxima during the summer time, where both maxima were observed in Alert as well as in Barrow (Quinn et al., 2007). The later maximum occurs in July and August and is due to the local productivity of phytoplankton while the surface water is free of ice. The earlier maximum that occurs around the time of our measurements,

5   can be associated with long-range transport from marine source regions from the North Pacific (Li et al., 1993). This fits well to the source region we found for the air mass of $PNSD_{c3}$ and can explain the presence of the Aitken mode particles. The minimum between the Aitken and the accumulation mode can be explained by previous cloud processing during which further material was added to activated droplets via aqueous phase oxidation. After the evaporation of cloud droplets this process creates the bi-modal $PNSD$ with the Hoppel-minimum (Hoppel et al., 1994). In our case the Hoppel-minimum can be found

10   at around 90 nm. **While cloud processing is a well known process for gaining particulate matter and growing particles to larger sizes, particles can also grow by generation of particulate matter directly from the gas phase as described recently for Arctic conditions in e.g., Willis et al. (2016), Burkart et al. (2017b) and Collins et al. (2017). The observed minimum in the PNSD occurs when new particle formation takes place, either by adding small particles to an already aged air mass or by mixing of different air masses with one air mass containing aged and the other one newly formed**

15   **particles, where one could come from aloft. It should also be mentioned that it was recently described in Gunsch et al. (2017) and Kolesar et al. (2017), that emissions from Prudhoe Bay oil field, which is located at the northern shore of Alaska roughly 700 km west of our measurement location, influenced Arctic PNDSs by adding both high concentrations of small particles and particulate mass to larger particles. Summarizing there is a number of reasons that can add to the observed bi-modality of the size distribution, but small, comparably newly formed particles will make up the observed**

20   **Aitken mode in all cases. Sources of the precursor gases forming these particles will differ from spring to summer, as mentioned above (Li et al., 1993).**

[revised manuscript text omitted]

AMAP: AMAP Assessment 2006: Acidifying Pollutants, Arctic Haze, and Acidification in the Arctic., Arctic Monitoring and Assessment Programme (AMAP), 2006.

Andreae, M. O. and Rosenfeld, D.: Aerosol–cloud–precipitation interactions. Part 1. The nature and sources of cloud-active aerosols, Earth-Sci. Rev., 89, 13 – 41, https://doi.org/http://dx.doi.org/10.1016/j.earscirev.2008.03.001, http://www.sciencedirect.com/science/article/pii/S0012825208000317, 2008.

Ashbaugh, L. L., Malm, W. C., and Sadeh, W. Z.: A residence time probability analysis of sulfur concentrations at grand Canyon National Park, Atmos. Environ., 19, 1263 – 1270, https://doi.org/10.1016/0004-6981(85)90256-2, http://www.sciencedirect.com/science/article/pii/0004698185902562, 1985.

Asmi, E., Kondratyev, V., Brus, D., Laurila, T., Lihavainen, H., Backman, J., Vakkari, V., Aurela, M., Hatakka, J., Viisanen, Y., Uttal, T., Ivakhov, V., and Makshtas, A.: Aerosol size distribution seasonal characteristics measured in Tiksi, Russian Arctic, Atmos. Chem. Phys., 16, 1271–1287, https://doi.org/10.5194/acp-16-1271-2016, <GotoISI>://WOS:000371284100005, 2016.

Behrenfeld, U., Krejci, R., Ström, J., and Stohl, A.: Chemical properties of Arctic aerosol particles collected at the Zeppelin station during the aerosol transition period in May and June of 2004, Tellus B, 60, 405–415, https://doi.org/10.1111/j.1600-0889.2008.00349.x, http://dx.doi.org/10.1111/j.1600-0889.2008.00349.x, 2008.

Browse, J., Carslaw, K. S., Arnold, S. R., Pringle, K., and Boucher, O.: The scavenging processes controlling the seasonal cycle in Arctic sulphate and black carbon aerosol, Atmos. Chem. Phys., 12, 6775–6798, https://doi.org/10.5194/acp-12-6775-2012, <GotoISI>://WOS:000308287000006, 2012.

Burkart, J., Willis, M. D., Bozem, H., Thomas, J. L., Law, K., Hoor, P., Aliabadi, A. A., Kollner, F., Schneiders, J., Herber, A., Abbatt, J. P. D., and Leaitch, W. R.: Summertime observations of elevated levels of ultrafine particles in the high Arctic marine boundary layer, Atmos. Chem. Phys., 17, 5515–5535, https://doi.org/10.5194/acp-17-5515-2017, <GotoISI>://WOS:000400506500002, 2017a.

Burkart, J., Hodshire, A. L., Mungall, E. L., Pierce, J. R., Collins, D. B., Ladino, L. A., Lee, A. K. Y., Irish, V., Wentzell, J. J. B., Liggio, J., Papakyriakou, T., Murphy, J., and Abbatt, J.: Organic condensation and particle growth to CCN sizes in the summertime marine Arctic is driven by materials more semivolatile than at continental sites, Geophys. Res. Lett., 44, https://doi.org/10.1002/2017GL075671, 2017b.

Collins, D. B., Burkart, J., Chang, R. Y. W., Lizotte, M., Boivin-Rioux, A., Blais, M., Mungall, E. L., Boyer, M., Irish, V. E., Masse, G., Kunkel, D., Tremblay, J. E., Papakyriakou, T., Bertram, A. K., Bozem, H., Gosselin, M., Levasseur, M., and Abbatt, J. P. D.: Frequent ultrafine particle formation and growth in Canadian Arctic marine and coastal environments, Atmos. Chem. Phys., 17, 13 119–13 138, https://doi.org/10.5194/acp-17-13119-2017, <GotoISI>://WOS:000414601800002, 2017.

Croft, B., Martin, R. V., Leaitch, W. R., Tunved, P., Breider, T. J., D'Andrea, S. D., and Pierce, J. R.: Processes controlling the annual cycle of Arctic aerosol number and size distributions, Atmos. Chem. Phys., 16, 3665–3682, https://doi.org/10.5194/acp-16-3665-2016, <GotoISI>://WOS:000374702300001, 2016a.

Croft, B., Wentworth, G. R., Martin, R. V., Leaitch, W. R., Murphy, J. G., Murphy, B. N., Kodros, J. K., Abbatt, J. P. D., and Pierce, J. R.: Contribution of Arctic seabird-colony ammonia to atmospheric particles and cloud-albedo radiative effect, Nat. Commun., 7, https://doi.org/10.1038/ncomms13444, <GotoISI>://WOS:000387972500001, 2016b.

Dall'Osto, M., Beddows, D. C. S., Tunved, P., Krejci, R., Ström, J., Hansson, H.-C., Yoon, Y. J., Park, K.-T., Becagli, S., Udisti, R., Onasch, T., O'Dowd, C. D., Simó, R., and Harrison, R. M.: Arctic sea ice melt leads to atmospheric new particle formation, Scientific Reports, 7, https://doi.org/10.1038/s41598-017-03328-1, https://doi.org/10.1038/s41598-017-03328-1, 2017a.

Dall'Osto, M., Ovadnevaite, J., Paglione, M., Beddows, D. C., Ceburnis, D., Cree, C., Cortés, P., Zamanillo, M., Nunes, S. O., Pérez, G. L., et al.: Antarctic sea ice region as a source of biogenic organic nitrogen in aerosols, Sci. Rep., 7, 6047, 2017b.

Engvall, A.-C., Krejci, R., Ström, J., Treffeisen, R., Scheele, R., Hermansen, O., and Paatero, J.: Changes in aerosol properties during spring-summer period in the Arctic troposphere, Atmos. Chem. Phys., 8, 445–462, https://doi.org/10.5194/acp-8-445-2008, http://www.atmos-chem-phys.net/8/445/2008/, 2008.

Fleming, Z. L., Monks, P. S., and Manning, A. J.: Review: Untangling the influence of air-mass history in interpreting observed atmospheric composition, Atmos. Res., 104-105, 1 – 39, https://doi.org/10.1016/j.atmosres.2011.09.009, http://www.sciencedirect.com/science/article/pii/S0169809511002948, 2012.

Freud, E., Krejci, R., Tunved, P., Leaitch, R., Nguyen, Q. T., Massling, A., Skov, H., and Barrie, L.: Pan-Arctic aerosol number size distributions: seasonality and transport patterns, Atmos. Chem. Phys., 17, 8101–8128, https://doi.org/10.5194/acp-17-8101-2017, <GotoISI>://WOS:000404773700005, 2017.

Garrett, T. J., Radke, L. F., and Hobbs, P. V.: Aerosol Effects on Cloud Emissivity and Surface Longwave Heating in the Arctic, J. Atmos. Sci., 59, 769–778, https://doi.org/10.1175/1520-0469(2002)059<0769:AEOCEA>2.0.CO;2, http://dx.doi.org/10.1175/1520-0469(2002)059<0769:AEOCEA>2.0.CO;2, 2002.

Gunsch, M. J., Kirpes, R. M., Kolesar, K. R., Barrett, T. E., China, S., Sheesley, R. J., Laskin, A., Wiedensohler, A., Tuch, T., and Pratt, K. A.: Contributions of transported Prudhoe Bay oil field emissions to the aerosol population in Utqiagvik, Alaska, Atmos. Chem. Phys., 17, 10 879–10 892, https://doi.org/10.5194/acp-17-10879-2017, <GotoISI>://WOS:000410640900001, 2017.

Gysel, M. and Stratmann, F.: WP3 - NA3: In-situ chemical, physical and optical properties of aerosols, Deliverable D3.11: Standardized protocol for CCN measurements, Tech. rep., http://www.actris.net/Publications/ACTRISQualityStandards/tabid/11271/language/en-GB/Default.aspx, 2013.

Heintzenberg, J.: Particle size distribution and optical properties of Arctic haze, Tellus, 32, 251–260, https://doi.org/10.1111/j.2153-3490.1980.tb00952.x, http://dx.doi.org/10.1111/j.2153-3490.1980.tb00952.x, 1980.

Hopke, P. K.: Review of receptor modeling methods for source apportionment, J. Air Waste Manag. Assoc., 66, 237–259, https://doi.org/10.1080/10962247.2016.1140693, http://dx.doi.org/10.1080/10962247.2016.1140693, pMID: 26756961, 2016.

Hoppel, W. A., Frick, G. M., Fitzgerald, J. W., and Larson, R. E.: Marine boundary layer measurements of new particle formation and the effects nonprecipitating clouds have on aerosol size distribution, J. Geophys. Res.: Atmos., 99, 14 443–14 459, https://doi.org/10.1029/94JD00797, http://dx.doi.org/10.1029/94JD00797, 1994.

Iversen, T. and Joranger, E.: Arctic air pollution and large scale atmospheric flows, Atmos. Environ., 19, 2099 – 2108, https://doi.org/http://dx.doi.org/10.1016/0004-6981(85)90117-9, http://www.sciencedirect.com/science/article/pii/0004698185901179, 1985.

Jacob, D. J., Crawford, J. H., Maring, H., Clarke, A. D., Dibb, J. E., Emmons, L. K., Ferrare, R. A., Hostetler, C. A., Russell, P. B., Singh, H. B., Thompson, A. M., Shaw, G. E., McCauley, E., Pederson, J. R., and Fisher, J. A.: The Arctic Research of the Composition of the Troposphere from Aircraft and Satellites (ARCTAS) mission: design, execution, and first results, Atmos. Chem. Phys., 10, 5191–5212, https://doi.org/10.5194/acp-10-5191-2010, http://www.atmos-chem-phys.net/10/5191/2010/, 2010.

Kammermann, L., Gysel, M., Weingartner, E., Herich, H., Cziczo, D. J., Holst, T., Svenningsson, B., Arneth, A., and Baltensperger, U.: Subarctic atmospheric aerosol composition: 3. Measured and modeled properties of cloud condensation nuclei, J. Geophys. Res.: Atmos., 115, https://doi.org/10.1029/2009JD012447, http://dx.doi.org/10.1029/2009JD012447, d04202, 2010.

Keegan, K. M., Albert, M. R., McConnell, J. R., and Baker, I.: Climate change and forest fires synergistically drive widespread melt events of the Greenland Ice Sheet, Proc. Natl. Acad. Sci., 111, 7964–7967, https://doi.org/10.1073/pnas.1405397111, <GotoISI>://WOS: 000336687900035, 2014.

Köhler, H.: The nucleus in and the growth of hygroscopic droplets, Trans. Faraday Soc., 32, 1152–1161, https://doi.org/10.1039/TF9363201152, http://dx.doi.org/10.1039/TF9363201152, 1936.

Kolesar, K. R., Cellini, J., Peterson, P. K., Jefferson, A., Tuch, T., Birmili, W., Wiedensohler, A., and Pratt, K. A.: Effect of Prudhoe Bay emissions on atmospheric aerosol growth events observed in Utqiagvik (Barrow), Alaska, Atmos. Environ., 152, 146–155, https://doi.org/10.1016/j.atmosenv.2016.12.019, <GotoISI>://WOS:000394400000013, 2017.

Kristensen, T. B., Müller, T., Kandler, K., Benker, N., Hartmann, M., Prospero, J. M., Wiedensohler, A., and Stratmann, F.: Properties of cloud condensation nuclei (CCN) in the trade wind marine boundary layer of the western North Atlantic, Atmos. Chem. Phys., 16, 2675–2688, https://doi.org/10.5194/acp-16-2675-2016, https://www.atmos-chem-phys.net/16/2675/2016/, 2016.

Lathem, T. L., Beyersdorf, A. J., Thornhill, K. L., Winstead, E. L., Cubison, M. J., Hecobian, A., Jimenez, J. L., Weber, R. J., Anderson, B. E., and Nenes, A.: Analysis of CCN activity of Arctic aerosol and Canadian biomass burning during summer 2008, Atmos. Chem. Phys., 13, 2735–2756, https://doi.org/10.5194/acp-13-2735-2013, http://www.atmos-chem-phys.net/13/2735/2013/, 2013.

Law, K. S. and Stohl, A.: Arctic Air Pollution: Origins and Impacts, Science, 315, 1537–1540, https://doi.org/10.1126/science.1137695, http://science.sciencemag.org/content/315/5818/1537, 2007.

Leaitch, R. W., Sharma, S., Huang, L., Toom-Sauntry, D., Chivulescu, A., Macdonald, A. M., von Salzen, K., Pierce, J. R., Bertram, A. K., Schroder, J. C., Shantz, N. C., Chang, R. Y.-W., and Norman, A.-L.: Dimethyl sulfide control of the clean summertime Arctic aerosol and cloud, Elem. Sci. Anthr., 1, https://doi.org/10.12952/journal.elementa.000017, 2013.

Leaitch, W. R., Korolev, A., Aliabadi, A. A., Burkart, J., Willis, M. D., Abbatt, J. P. D., Bozem, H., Hoor, P., Kollnr, F., Schneider, J., Herber, A., Konrad, C., and Brauner, R.: Effects of 20-100nm particles on liquid clouds in the clean summertime Arctic, Atmos. Chem. Phys., 16, 11 107–11 124, https://doi.org/10.5194/acp-16-11107-2016, <GotoISI>://WOS:000384006600001, 2016.

[revised manuscript text omitted]